# Multimodal electrophysiological analyses reveal that reduced synaptic excitatory neurotransmission underlies seizures in a model of NMDAR antibody-mediated encephalitis

Sukhvir K. Wright [1,2 ✉], Richard E. Rosch [3,4,5 ✉], Max A. Wilson [1], Manoj A. Upadhya[1], Divya R. Dhangar [1], Charlie Clarke-Bland[1], Tamara T. Wahid[1], Sumanta Barman[6], Norbert Goebels[6], Jakob Kreye[7,8,9], Harald Prüss [7,8], Leslie Jacobson[10], Danielle S. Bassett [5,11], Angela Vincent[10], Stuart D. Greenhill [1] & Gavin L. Woodhall [1 ✉]

Seizures are a prominent feature in N-Methyl-D-Aspartate receptor antibody (NMDAR antibody) encephalitis, a distinct neuro-immunological disorder in which specific human autoantibodies bind and crosslink the surface of NMDAR proteins thereby causing internalization and a state of NMDAR hypofunction. To further understand ictogenesis in this disorder, and to test a potential treatment compound, we developed an NMDAR antibody mediated rat seizure model that displays spontaneous epileptiform activity in vivo and in vitro. Using a combination of electrophysiological and dynamic causal modelling techniques we show that, contrary to expectation, reduction of synaptic excitatory, but not inhibitory, neurotransmission underlies the ictal events through alterations in the dynamical behaviour of microcircuits in brain tissue. Moreover, in vitro application of a neurosteroid, pregnenolone sulphate, that upregulates NMDARs, reduced established ictal activity. This proof-of-concept study highlights the complexity of circuit disturbances that may lead to seizures and the potential use of receptor-specific treatments in antibody-mediated seizures and epilepsy.

[1] Institute of Health and Neurodevelopment, College of Health and Life Sciences, Aston University, Birmingham, UK. [2] Department of Paediatric Neurology, The Birmingham Women's and Children's Hospital NHS Foundation Trust, Birmingham, UK. [3] MRC Centre for Neurodevelopmental Disorders, King's College London, London, UK. [4] Department of Paediatric Neurology, Great Ormond Street Hospital for Children NHS Foundation Trust, London, UK. [5] Department of Bioengineering, University of Pennsylvania, Philadelphia, PA, USA. [6] Department of Neurology, Medical Faculty, Heinrich Heine University Düsseldorf, Düsseldorf, Germany. [7] German Center for Neurodegenerative Diseases (DZNE) Berlin, Berlin, Germany. [8] Department of Neurology and Experimental Neurology, Charité – Universitätsmedizin Berlin, Berlin, Germany. [9] Department of Pediatric Neurology, Charité – Universitätsmedizin Berlin, Berlin, Germany. [10] Nuffield Department of Clinical Neurosciences, Oxford University, Oxford, UK. [11] Santa Fe Institute, Santa Fe, NM, USA. ✉email: s.wright5@aston.ac.uk; richard.rosch@kcl.ac.uk; g.l.woodhall@aston.ac.uk

Both in vivo and in vitro studies have clearly established that the pathogenic action of antibodies in N-methyl-D-aspartate receptor (NMDAR) antibody encephalitis is to bind to the surface of the NMDARs, cross-link the proteins and cause NMDAR internalization, resulting in a state of NMDAR hypofunction[1–3]. As a result, the mainstay of treatment is immunotherapy that aims to reduce the levels of circulating neuronal autoantibodies or halt their production[4,5]. These treatments can be slow to work, and carry the risk of harmful side-effects as the healthy immune system is also compromised[6]. It is clear that the reduction of NMDARs in NMDAR antibody encephalitis causes a myriad of severe neurological symptoms, and that reversal of this pathological effect in the acute and even chronic stages of the disease has the potential to improve clinical outcomes[7–9]. However, as yet, there are no immune-sparing treatments that ameliorate the synaptic and circuit effects (psychosis, ictogenesis) caused by the NMDAR hypofunction[10,11]. The therapeutic potential of a receptor-specific treatment could be particularly beneficial in children who are affected during crucial developmental stages, marked by neuronal network plasticity including that of the NMDAR subunits[12,13]. Indeed, studies have found that after recovery children have persistent cognitive problems and fatigue, resulting in lower academic achievement and poorer quality of life[14,15].

While neuropsychiatric features are the most common presenting symptom of NMDAR antibody encephalitis, acute symptomatic seizures and encephalopathic electroencephalogram (EEG) changes are essential clinical and investigative features in patients of all ages[16]. Seizures present as part of the disease course in >80% of patients and EEG changes are seen in >95%, more commonly than Magnetic Resonance Imaging (MRI) brain changes, as noted repeatedly in cohort and case studies[4,5,17]. Despite the symptom predominance of seizures in humans, producing reliable and consistent spontaneous seizure animal models of this disorder using passive transfer of human-derived antibodies is challenging[18–20]. In our previous passive transfer mouse model, we demonstrated increased in vivo seizure susceptibility (and therefore network hyperexcitability) 48 h after intracerebroventricular NMDAR antibody injection compared to controls, however spontaneous seizures were not seen[18,21].

Importantly, seizures within the context of NMDAR antibody encephalitis do not respond well to standard anti-seizure medications, as seen in other forms of antibody-mediated encephalitis[22–24]. Given the specific involvement of the excitatory NMDAR, it is tempting to speculate that the pathophysiology of these epilepsies diverges from a simple 'excitation-inhibition' imbalance and that an NMDAR-specific treatment may be more effective in minimizing acute symptoms such as seizures, and could potentially be given alongside immunotherapy to achieve more rapid symptomatic relief.

This study aimed to understand the causative changes in synaptic physiology contributing to NMDAR antibody-mediated seizures, and to explore the unmet clinical need for receptor-specific treatment. We developed a juvenile Wistar rat model of NMDAR antibody encephalitis, in which NMDAR antibodies derived from human patients were used to induce a robust and repeatable seizure phenotype. Combining in vitro, in vivo, and in silico approaches we have demonstrated that, counterintuitively, the epileptic dynamics emerge from reductions in excitatory neurotransmission caused by NMDAR antibodies, and provide proof-of-concept in vitro evidence of a NMDAR specific neurosteroid rescue treatment both in the rodent model, and in human epileptic brain tissue.

## Results

### NMDAR antibodies bind specifically to Wistar rat brain tissue.
All three NMDAR antibody preparations derived from patients with the disease bound specifically to rodent hippocampus (Supplementary Fig. 1) confirming previous studies in mice[3,8,18]. The florescence intensities of NMDAR antibody injected and infused brain slices were significantly higher than those measured after injection or infusion of control human IgG or non-brain reactive monoclonal antibodies (mGO53, 12D7) (Supplementary Fig. 1).

### NMDAR antibodies cause spontaneous epileptiform activity in vitro.
In order to characterize NMDAR antibody induced abnormalities of synaptic transmission and circuit dysfunction, we first examined the effects of the antibody preparations on brain-slices in vitro. At 48 h after intracerebroventricular (ICV) injection, juvenile rats injected with NMDAR antibodies exhibited sustained and repetitive hyperexcitable behaviour including myoclonic twitches, jerks and jumps (Supplementary Movie 1). Animals were sacrificed at this time-point and brain slices were prepared for in vitro electrophysiology studies. These hippocampal local field potential (LFP) recordings showed significantly more frequent spontaneous epileptiform activity in NMDAR antibody injected slices compared to slices prepared from control antibody injected animals (Fig. 1a–d). In the NMDAR antibody injected animals the number of interictal events was higher in the CA3 hippocampal region compared to CA1, with reduced interevent interval (IEI) between spikes (Fig. 1e, f).

To determine the synaptic changes contributing to this apparent circuit hyperexcitability, whole-cell patch clamp recordings were made from pyramidal cells in the CA3 region of the hippocampus (Fig. 2a; cell morphology confirmed on post-hoc analysis, Supplementary Fig. 1). There was a reduction in the frequency and amplitude of spontaneous excitatory post-synaptic current (sEPSC) recordings (Fig. 2b–d) and a significant shift towards a faster decay time indicating loss of NMDAR-mediated events (Fig. 2e, f). By contrast, no changes were seen in the frequency or amplitude of spontaneous inhibitory post-synaptic currents (sIPSCS) or miniature IPSCs (Fig. 2g–l). Additionally, evaluation of the excitability and intrinsic properties of recorded CA3 pyramidal neurons showed them to be hyperexcitable (Fig. 2m–p). Together these results indicate that local hippocampal networks are hyperexcitable when recorded in vitro 48 h after a single ICV injection of NMDAR antibodies in vivo. This circuit hyperexcitability is paradoxically associated with a reduction of excitatory neurotransmission in CA3 induced by the NMDAR antibodies with compensatory changes in the intrinsic properties of recorded CA3 pyramidal cells.

### NMDAR antibodies cause spontaneous epileptic seizures in vivo.
To demonstrate ictogenesis in vivo, 7-day osmotic pumps were used to deliver NMDAR antibodies or control antibodies into the lateral cerebral ventricles of juvenile Wistar rats ($n = 6$ animals for each group). Following the observations above, a depth electrode was placed in the CA3 region. Spontaneous epileptiform events were observed in the NMDAR antibody infused animals in vivo and evident in the EEG tracings (Fig. 3a, Supplementary Movies 2–4). A significant increase in EEG coastline of the NMDAR antibody animals compared to control antibody treated reflected this increase in occurrence of epileptiform high amplitude events (Fig. 3b). Using automated seizure detection[18,25], we observed a significant increase in the number of ictal events per hour in the NMDAR antibody animals compared to controls. The maximum ictal event frequency was seen in NMDAR antibody animals during Day 2 of the infusion, but it remained significantly elevated compared to controls until the end of the recording and infusion (Fig. 3c–e). The EEG of NMDAR antibody animals exhibited increased power in all

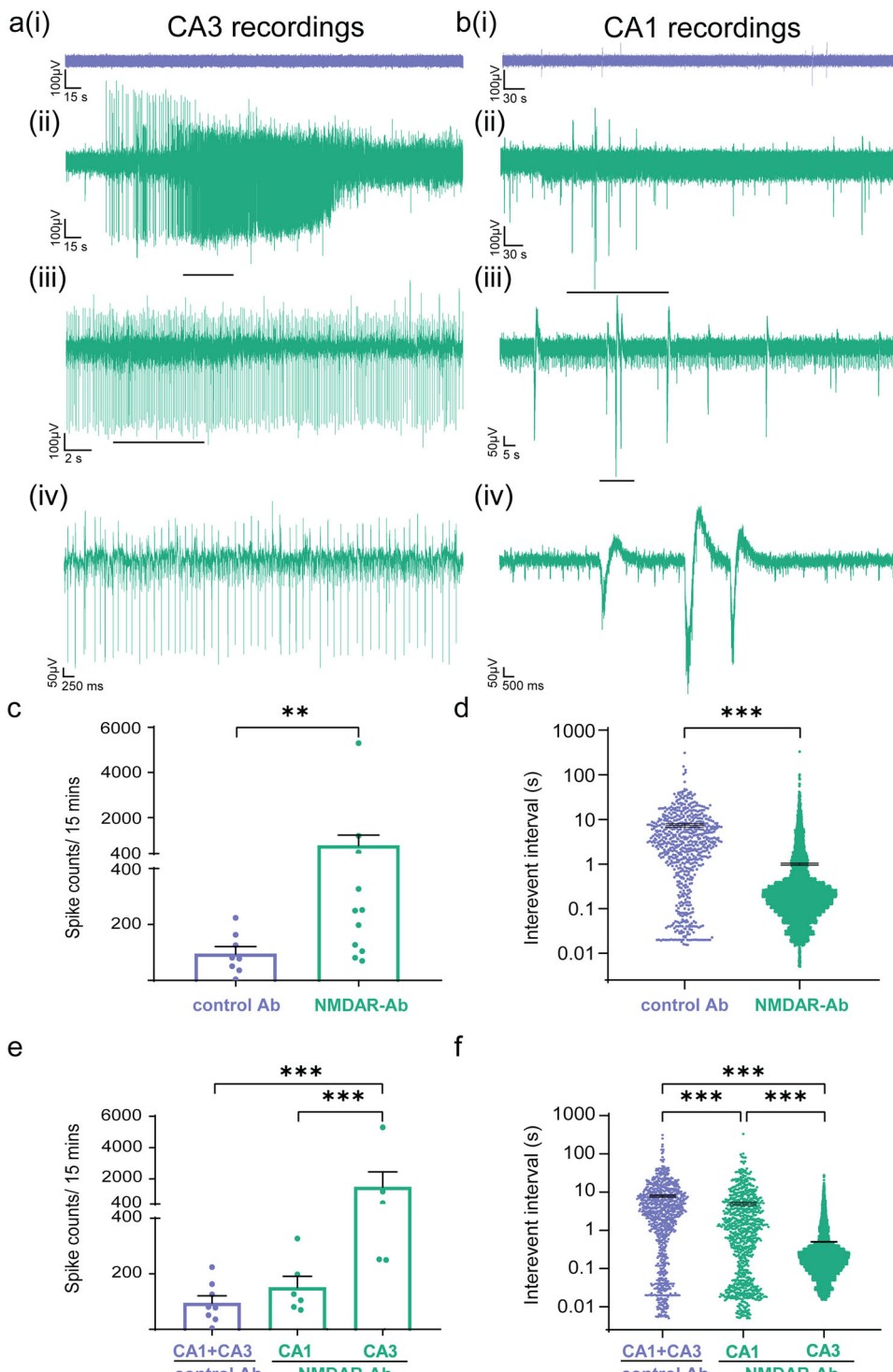

**Fig. 1 Hippocampal local field potential recordings in vitro 48 h after intracerebroventricular injection of NMDAR antibodies show spontaneous epileptic activity, highest in the CA3 region. a** Example traces of local field potential brain slice recordings from control (**a**i) and NMDAR antibody (**a**ii–iv) injected animals after 48 h from the CA3 region. Scale bar 100 μV vs. 15 s (i and ii), 100 μV vs. 2 s (iii) and 50 μV vs. 250 msec (iv). **b** Example traces of local field potential brain slice recordings from control (**b**i) and NMDAR antibody (**b**ii–iv) injected animals after 48 h from the CA1 region. Scale bar 100 μV vs. 30 s (I and ii), 50 μV vs. 5 s (iii) and 50 μV vs. 500 msec (iv). **c** The number of spikes (interictal events) per 15 min in the NMDAR antibody slices compared to controls (Control ($n = 8$ brain slices) vs. NMDA ($n = 11$ brain slices); **$p = 0.0065$, Mann–Whitney). **d** The intervent interval (IEI) in control Ab slices compared to NMDAR antibody injected slices (Control ($n = 8$ brain slices) vs. NMDA ($n = 11$ brain slices); ***$p < 0.001$, Mann–Whitney). **e** The number of spikes (interictal events) in the CA3 region of the NMDAR antibody hippocampal slices as compared to CA1 (NMDA CA1 ($n = 6$ brain slices) vs. NMDA CA3 ($n = 5$ brain slices); ***$p < 0.001$, Mann–Whitney). **f** Comparison of IEI between the CA3 and CA1 region in NMDAR antibody hippocampal slices (NMDA CA1 ($n = 6$ brain slices) vs. NMDA CA3 ($n = 5$ brain slices); ***$p < 0.001$, Mann–Whitney). Measurements expressed as mean ± standard error of the mean (SEM).

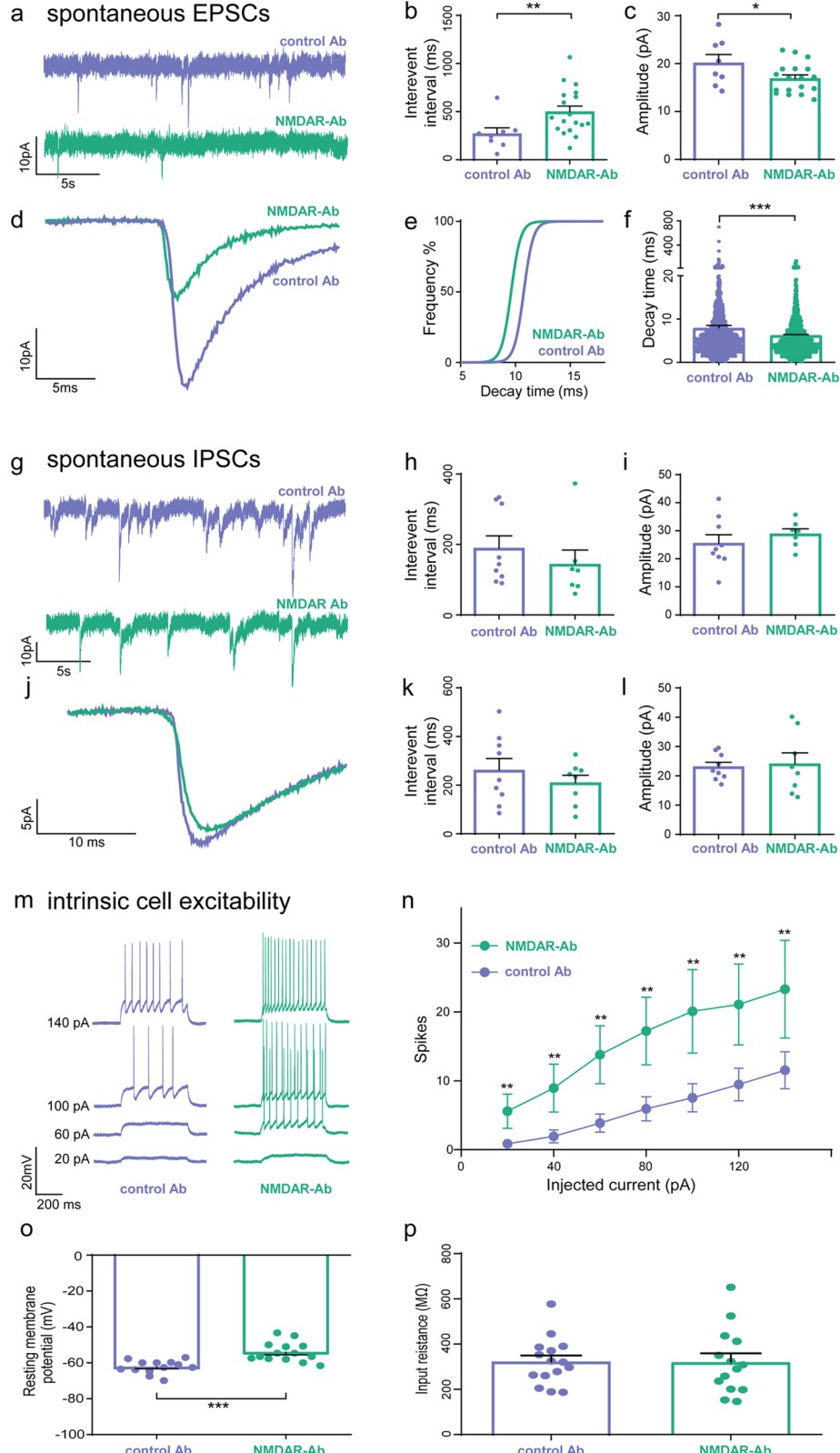

frequency bands measured compared to control antibody animals (1–4 Hz, 4–8 Hz, 8–12 Hz, 12–30 Hz, 30–50 Hz, 50–70 Hz, 70–120 Hz, and 120–180 Hz; $p = <0.0001$, Mann–Whitney). The greatest increase in power, seen in the 12–30 Hz range (676 ± 151 vs 3195 ± 102, $n = 6$; $p = <0.0001$, Mann–Whitney), was similar to the typical excessive beta activity observed in EEGs of NMDAR antibody patients (14–20 Hz)[26].

To confirm that these findings were concordant with the in vitro findings at 48 h, we recorded LFPs in hippocampal brain slices at the completion of the recording experiments (ranging from day 7 to 14). In line with the spontaneous epileptiform activity seen in vivo, there was a considerable increase in the number of interictal events and a reduced IEI of ictal spikes in NMDAR antibody infused animals compared to controls (Fig. 3f, g). CA3

**Fig. 2 Whole-cell patch-clamp recordings from hippocampal CA3 pyramidal cells in vitro 48 h after intracerebroventricular injection of NMDAR antibodies show a reduction in excitatory but not inhibitory synaptic neurotransmission and intrinsic hyperexcitability. a** Representative spontaneous EPSC (sEPSC) whole-cell patch clamp recordings 48 h after injection with NMDAR antibodies or control antibodies. Scale bar 10 pA vs. 5 s. **b** The interevent interval (IEI) of sEPSCs recorded from CA3 pyramidal cells in NMDAR antibody injected rodent slices ($n = 18$ cells) compared to control Ab ($n = 8$ cells) injected slices (**$p = 0.003$, Mann–Whitney). **c** The amplitude of the sEPSCs in CA3 pyramidal cells recorded from NMDAR antibody injected slices ($n = 18$ cells) compared to controls ($n = 8$ cells) (*$p = 0.04$; Mann–Whitney). **d** Representative average sEPSCs recorded from NMDAR antibody injected slices as compared to control antibody injected slices. Scale bar 10 pA vs. 10 ms. **e** Cumulative frequency plot showing reduced decay time of sEPSCs recorded from NMDAR antibody slices as compared to the controls ($n = 7$ cells in each group). **f** Decay time of the sEPSCs in the NMDAR antibody injected rodent slices as compared to the controls ($n = 7$ cells in each group; $p = <0.001$, Mann–Whitney). **g** Representative whole-cell patch clamp recording of spontaneous IPSC (sIPSC) from CA3 pyramidal cells after injection of NMDAR- or control antibodies. Scale bar 10 pA vs. 5 s. **h** The intervent interval of the sIPSCs recorded from CA3 pyramidal cells in control and NMDAR antibody injected slices ($n = 6$ cells in each group; $p = 0.156$, Mann–Whitney). **i** The amplitude of sIPSCS recorded from the control and NMDAR antibody injected slices ($n = 6$ cells in each group; $p = 0.15$, Mann–Whitney). **j** Representative average sIPSCs recorded from NMDAR antibody injected slices as compared to control antibody injected slices. Scale bar 5 pA vs. 10 ms. **k** The interevent interval of miniature IPSCs recorded (in the presence of TTX 1 μM) from pyramidal cells in CA3 in control and NMDAR antibody injected slices ($n = 6$ cells in each group; $p = 0.28$, Mann–Whitney). **l** The amplitude of miniature IPSCs recorded (in the presence of TTX 1 μM) from pyramidal cells in CA3 in control and NMDAR antibody injected slices ($n = 6$ cells in each group; $p = 0.41$, Mann–Whitney). **m** Representative traces of CA3 pyramidal cell responses during depolarising current steps in CA3 pyramidal cells from control and NMDAR antibody injected rodent slices. **n** Depolarising steps of different current intensities elicited significantly more spikes in the NMDAR antibody injected rodent slices ($n = 15$ cells) than in control antibody ($n = 15$ cells) treated slices (**$p = 0.005$, unpaired t-test). **o** The resting membrane potential was significantly depolarised in the NMDAR antibody treated ($n = 14$ cells) CA3 pyramidal cells compared to those treated with control antibody ($n = 13$ cells; ***$p = <0.0001$, Mann–Witney). **p** Input resistance was not significantly altered between the two conditions (NMDAR antibody, $n = 14$ cells vs. control antibody $n = 15$ cells; $p = 0.34$, Mann–Witney). Measurements expressed as mean (M) ± standard error of the mean (SEM).

hippocampal whole-cell patch clamp experiments in NMDAR antibody treated animals showed an even further increase in the IEI of sEPSCs (Fig. 3h), and further reduction of the amplitude (Fig. 3i) from that seen at 48 hours. Thus a reduction in excitatory neurotransmission was found to be coincident with the spontaneous epileptic events in both the 48 h and 7 day NMDAR antibody models.

**Reduced excitatory neurotransmission contributes to EEG changes.** We tested whether the in vitro changes in single neuron behaviour associated with NMDAR antibody contributed to in vivo (interictal) EEG patterns by using a computational model of microcircuit dynamics. Specifically, we tested the hypothesis that the microscale recordings can inform models of LFP dynamics using Bayesian model comparison of models with, and without prior information derived at the microscale. We first generated quantitative predictions of the population-level effect of NMDAR antibody based on in vitro differences between NMDAR antibody and control antibody exposed neurons measured using whole-cell patch clamp recordings (Fig. 4a, b). We then fitted a four-population neural mass model (the canonical microcircuit[27]) to EEG power spectral densities, first for control animals, then for NMDAR antibody animals under different empirical priors with model fits for the power spectral densities shown for the final, selected model in Fig. 5a. These empirical priors were based on posteriors from the control antibody treated animals and, for subsets of parameters, the quantitative predictions made from the in vitro recordings (see Methods). We then performed Bayesian model comparison of these models using a free energy approximation of the model evidence – briefly, we fitted models where different subsets of parameters were informed by the microscale priors to the same data. By comparing the model evidence across these model inversions, we can thus evaluate whether the microscale priors allow a more parsimonious explanation for the LFP data we have fitted. Given the uneven distribution of NMDAR in the cortical layers and across different neuronal populations, we also compare models where priors are applied only to different subsets of parameters, resulting in the model space of 22 models. This comparison provides two key insights: (i) the model equipped with the full set of quantitative predictions from the microscale provides a more parsimonious explanation for the observed EEG spectral differences between control antibody and NMDAR antibody, compared

to a null model without these microscale-derived predictions; and (ii) in a model space comparing models with only subsets of parameters informed by microscale-derived priors, the best model incorporates microscale-derived priors only for amplitude and population variance in one compartment of the canonical microcircuit model.

We then considered the overall winning model highlighted in Fig. 4c and used its posterior parameters as priors for fitting EEG power spectral densities for seizure recordings. This modelling approach thus provided us with a parameter set capturing the difference between interictal EEG spectra in NMDAR antibody and control antibody treated animals (Fig. 4d, middle panel), and between ictal EEG spectra and interictal EEG spectra in NMDAR antibody (Fig. 4d, bottom panel). In order to assess whether the two main effects of interest (changes caused by NMDAR antibody, and changes induced by seizures) can be separated in terms of their associated parameter changes, we performed a principal component analysis over three vectors containing the parameters fitted to control antibody, NMDAR antibody (interictal), and NMDAR antibody (ictal) states. Projecting the parameter combinations for each of the three states onto the first two principal components (Fig. 4e) demonstrates separability of the two effects with near orthogonal effects of NMDAR antibody and seizure onset. For subsequent analyses, we therefore considered these effects independently: (i) the synaptic parameter difference derived from *control antibody* and *NMDAR antibody (interictal)* conditions captures the main effect of NMDAR antibody infusion, which we know from the preceding results in a higher seizure likelihood, and is thus epileptogenic in the animals (hence, 'epileptogenesis parameters' for brevity). (ii) The synaptic parameter difference derived from the *NMDAR antibody (interictal)*, and the *NMDAR antibody (ictal)* conditions captures the main effect of transition into the seizure state, and thus represents ictogenesis (hence, 'ictogenesis parameters' for brevity).

**NMDAR antibodies push brain microcircuits into an unstable regime.** Our hypothesis was that NMDAR antibody alter brain dynamics in a way that makes neuronal circuits more susceptible to potentially seizure-inducing fluctuations in synaptic parameters. We tested this using the neuronal microcircuit models in simulation mode, allowing us to test any combination of

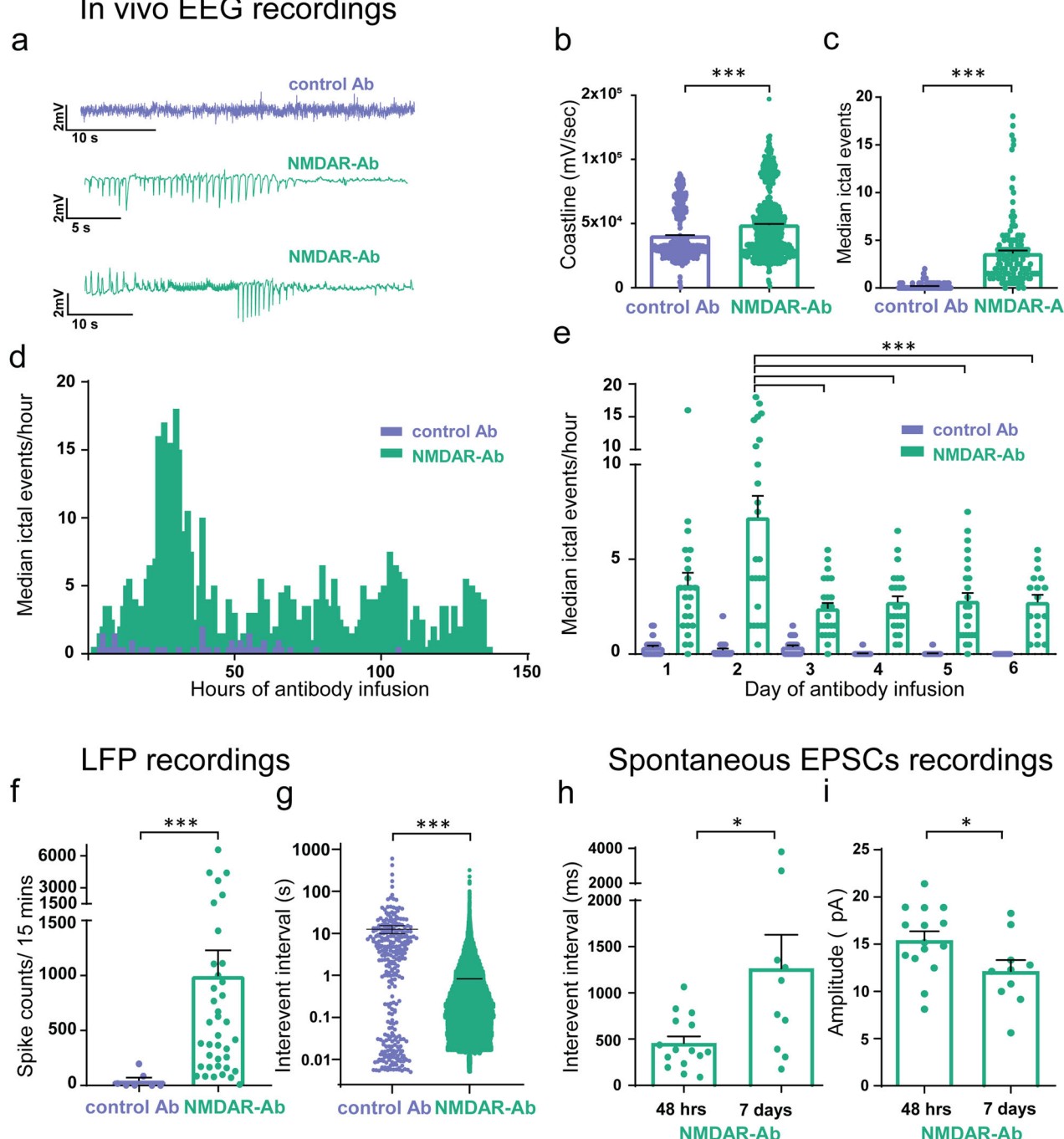

**Fig. 3 In vivo spontaneous epileptic EEG activity recorded in rodents infused with NMDAR antibodies. a** Example EEG recordings from experimental rodents using wireless EEG transmitters with osmotic pumps in situ delivering control or NMDAR antibodies. **b** EEG coastline length (calculated per second for the total 7 day recording period for each animal) in NMDAR antibody animals as compared to control animals ($n = 6$ animals in each group; ***$p < 0.0001$, Mann–Whitney). **c** The number of spontaneous ictal events/hour (mean median ± SEM) in the NMDAR antibody animals compared to controls (***$p < 0.001$; Mann–Whitney). **d** Graph illustrating median number of ictal events over time of control and NMDAR antibody infusion. **e** Daily comparison of ictal events in the NMDAR antibody and control antibody animals (***$p = <0.001$, two-way Anova with Bonferroni correction). **f** Ictal spike counts from hippocampal LFP measurements in brain slices prepared from NMDAR antibody and control antibody infused animals at the end of the in vivo EEG recording period (controls ($n = 7$) vs. NMDA ($n = 38$); ***$p < 0.001$, Mann–Whitney. **g** Interevent interval from hippocampal LFP measurements in brain slices prepared from NMDAR antibody and control antibody infused animals at the end of the in vivo EEG recording period (controls ($n = 7$ brain slices) ($12.56 ± 2.70$) vs. NMDA ($n = 38$ brain slices); ***$p < 0.001$; Mann–Whitney). **h** Interevent interval of sEPSCs recorded from putative CA3 cells in slices prepared from animals after 7 days of NMDAR antibody ($n = 15$ cells) as compared to the same measurements taken at 48 h after NMDAR antibody infusion ($n = 18$ cells) (*$p = 0.03$; Mann–Whitney). **i** Amplitude measurements of sEPSCs recorded from putative CA3 cells in slices prepared from animals after 7 days of NMDAR antibody ($n = 15$ cells) as compared to the same measurements taken at 48 h after NMDAR antibody infusion ($n = 18$ cells) (*$p = 0.03$; Mann–Whitney). Measurements expressed as mean (M) ± standard error of the mean (SEM).

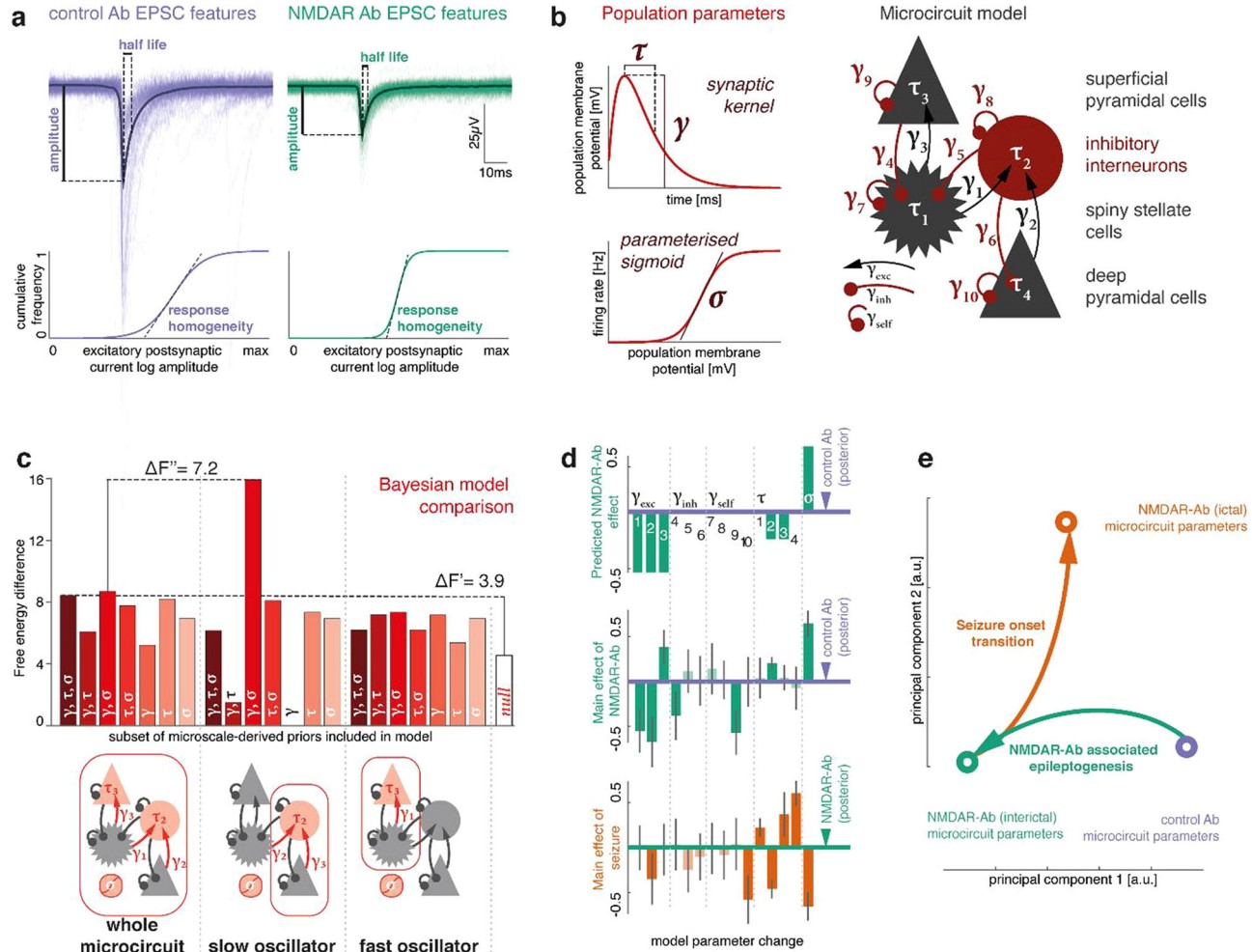

**Fig. 4 In vitro microscale disruptions of excitatory neurotransmission explain NMDAR antibody-associated in vivo neurophysiological changes.**
**a** In vitro recordings of sEPSCs were used to quantify differences between control antibody and NMDAR antibody related neurotransmission, including half-life and amplitude of EPSCs. We also estimated EPSC homogeneity by fitting a sigmoid curve to the cumulative frequency of EPSC amplitudes, where the slope at the midpoint increases with more homogeneous EPSC amplitude distribution. **b** Microscale data features map onto population parameters in a canonical microcircuit model consisting of two layered neuronal oscillator pairs with recurrent excitation-inhibition coupling. Key synaptic parameters of this model are population synaptic response amplitude, time constant and response sigmoid, which are affected by EPSC amplitude, time constant, and amplitude distribution, respectively. Using the dynamic causal modelling framework, we fit a single such microcircuit to local field potential recordings in control Ab and NMDAR antibody exposed mice. **c** Bayesian model comparison is shown for inverted models of NMDAR antibody (interictal) local field potential dynamics where subsets of the excitatory synaptic parameters were informed by priors derived from microscale recordings. There is decisive evidence that the full model, where all parameters had microscale-informed priors, provides a more parsimonious explanation for the data than the model without predictive priors (BF ~ ΔF' = 3.9). Of all tested models, the model containing predictive priors only for amplitude (γ) and population variance (σ), and only for the slow neuronal oscillator populations (inhibitory interneurons and deep pyramidal cells) performed best (BF ~ ΔF' = 3.9) with a posterior probability of >95% in the given model space. **d** Parameter differences between conditions are shown for control vs. interictal NMDAR antibody-associated dynamics (predicted NMDAR-Ab effect; main effect of NMDAR-Ab: estimated in the winning model), and seizure vs. interictal NMDAR antibody dynamics (main effect of seizure: estimated). Error bars indicate a Bayesian 95% credible interval. Darker shading parameters where the estimated difference exceeds the credible interval. **e** A principal component decomposition of the full parameter set across three states (control antibody, NMDAR antibody (interictal), and NMDAR antibody (seizure)) shows separability of the main effects of antibody (with most of the difference along the first principal component) and seizure onset (with most of the difference along the second principal component).

parameter values, including mixtures of the three empirically-fitted conditions. Specifically, we ran simulations across an 'epileptogenesis' × 'ictogenesis' parameter space. For this analysis, we consider the parameter differences between *control* and *NMDAR antibody (interictal)* condition the 'epileptogenesis' parameters since they capture the difference between brain dynamics in the non-epileptic control animals and the animals with NMDAR antibody induced spontaneous recurrent seizures. Similarly, we consider the parameter differences between *NMDAR antibody (interictal)* and *NMDAR (ictal)* the 'ictogenesis' parameters since

they capture the difference between brain dynamics in the interictal and ictal state. We derived initial parameterisations by fitting microcircuit models to the three empirically observed brain states as described above (control antibody, interictal NMDAR antibody and seizure NMDAR antibody; model fits shown in Fig. 5a). We then simulated linearly spaced intermediate steps between states along this 2-dimensional parameter space. At each point, we simulated the full spectrum, and classified it into one of the three empirically observed brain states by least mean squared difference between the simulated and observed spectra (Fig. 5b).

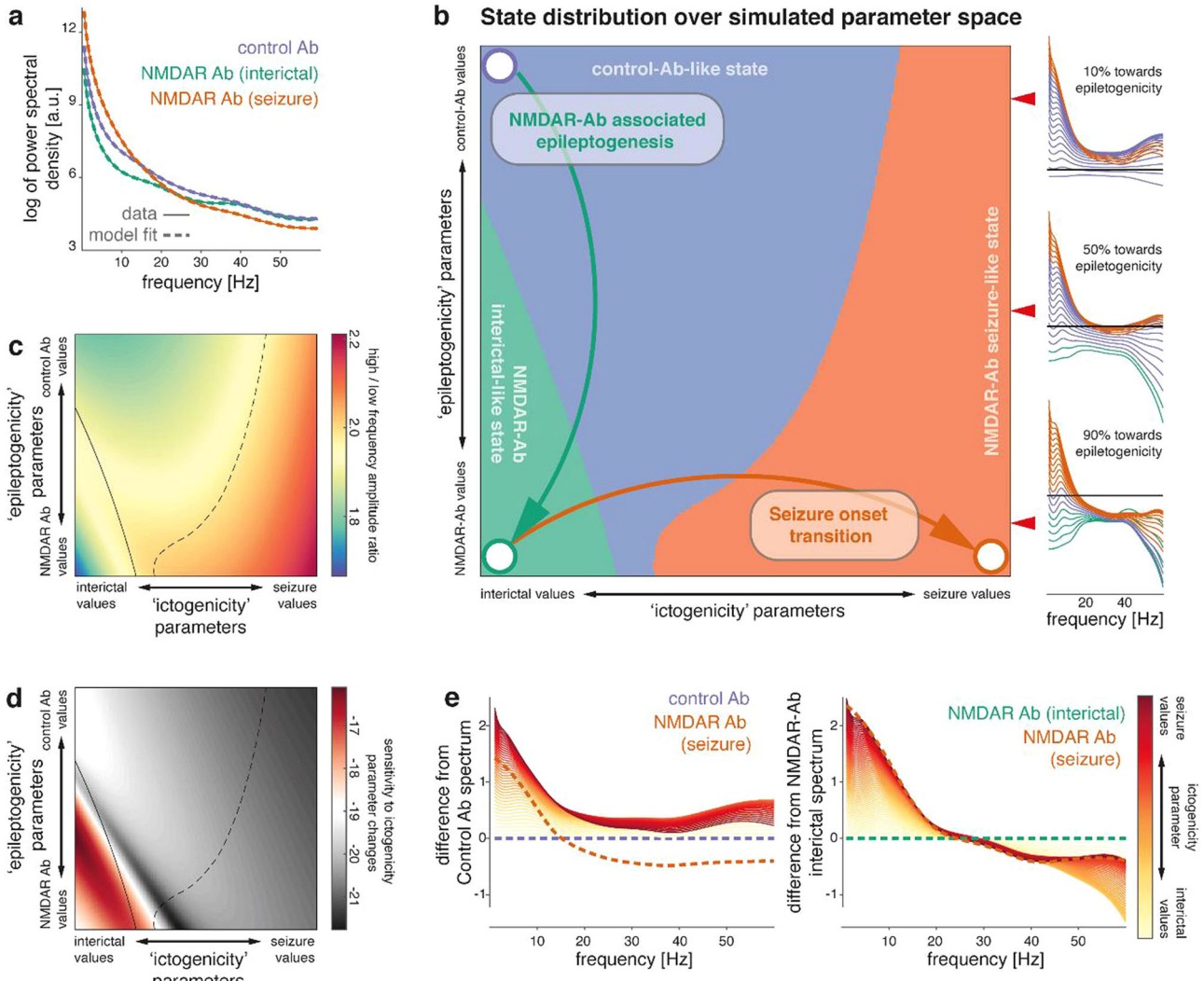

**Fig. 5 NMDAR antibody pushes neuronal microcircuits towards unstable regimes. a** Power spectra of LFP recordings of control antibody injected mice, and interictal and ictal states of NMDAR antibody injected mice are fitted with neural mass models of neuronal microcircuits. Dashed lines represent the model fit; solid lines represent the empirically measured spectra, indicating faithful reproduction of empirically measured spectral features in the fitted dynamic causal model. **b** Based on the parameters derived from the dynamic causal model fits (represented here by circles), parameter changes that capture NMDAR antibody associated alterations in microcircuitry ('epileptogenicity') and seizure onset transition ('ictogenicity') were identified (arrows). Linear combinations of these parameters are then used to simulate spectra at different locations in a two-dimensional parameter space along both 'epileptogenicity' on the y-axis, and 'ictogenicity' on the x-axis. At each point full spectra were generated, and classified as one of three possible states ('Control antibody-like', 'NMDAR antibody interictal-like', 'NMDAR antibody seizure-like') based on the smallest mean-squared difference to the three empirically measured spectra. Insets show example simulations (normalized to the 'Control antibody' spectrum, indicated by the black line) at three different values for 'epileptogenicity' indicated by the red arrow heads (10%, 50%, 90% towards full 'epileptogenicity' parameter change). Each line represents a simulation at different ictogenicity values spanning the full width of ictogenicity parameters, colour-coded by their classification. **c** Along the 'epileptogenicity' vs.'ictogenicity' parameter space, the ratio of high frequency (>20 Hz) and low-frequency (<8 Hz) log amplitude is shown with state separating boundaries from panel (**b**) overlaid. **d** This plot demonstrates sensitivity to changes in ictogenicity parameters. For each given 'epileptogenicity' (i.e. row in the parameter space plot), mean-squared differences between neighbouring values of 'ictogenicity' are shown along the log-scale (a.u.) colour axis. State separating boundaries from (**b**) are overlaid. There is considerable overlap between regions in parameter space, where the microcircuit is sensitive to perturbation, and the NMDAR antibody-like state identified in (**b**). **e** To demonstrate differences in sensitivity to 'ictogenicity' parameter changes, spectral differences from both control Ab parmeterisation and NMDAR Ab parameterisation are shown for increasing contribution of 'ictogenicity' parameters.

This map of parameter space shows that for microcircuits at control antibody parameterization, the transition to a putative seizure-like territory is further away on the 'ictogenesis' parameter axis than it is for microcircuits at the interictal NMDAR antibody parameterization. Each of the territories is characterised by particular spectral features, illustrated in Fig. 5c by a map of high frequency and low frequency amplitude ratios across the same parameter space.

To evaluate sensitivity of these microcircuits to changes in the ictogenicity parameters, we quantified the difference of each simulated spectrum to spectra simulated from similar, 'neighbouring' ictogenicity parameter values (Fig. 5d). This analysis revealed that the NMDAR antibody interictal state is most sensitive to small changes in ictogenicity parameters.

Spectral changes associated with the increasing effect of ictogenicity parameters are further shown in Fig. 5e, relative to

control antibody and interictal NMDAR antibody microcircuit parameterisations.

**Pregnenolone sulphate (PregS) rescues NMDAR antibody epileptic changes in vitro.** We hypothesised that increasing NMDAR expression could rescue the electrophysiological features associated with NMDAR antibody. The neurosteroid pregnenolone sulphate (PregS) has a membrane delimited action to increase the trafficking of functional NMDARs to the neuronal membrane[28]. Using whole-cell patch clamp recordings of sEPSCs from untreated Wistar rat hippocampal slices we showed that PregS significantly increased amplitude and frequency of sEPSCs; the increased frequency was reversed by application of the NMDAR antagonist MK801 confirming NMDAR mediated effect (Supplementary Fig. 2a–d). PregS reduced sIPSC IEI ($162 \pm 47.5$ vs $101 \pm 36.7$, $n = 9$ cells; $p = 0.03$, Wilcoxon paired rank test), but did not affect the amplitude ($22.81 \pm 2.5$ vs $30.90 \pm 3.2$, $n = 9$ cells; $p = 0.25$, Wilcoxon paired rank test) in control slices.

Next we tested whether these effects would rescue the NMDAR antibody induced abnormalities observed in vitro. We recorded LFP discharges in vitro before and after addition of PregS to brain slices from animals treated with NMDAR antibody or control antibody for 48 h or 7 days (Fig. 6a). The number of interictal/ spike events in NMDAR antibody treated slices was significantly reduced after application of PregS and there was a corresponding increase in IEI (Fig. 6b,c). By contrast, slices from rats treated with control antibody showed no significant reduction in interictal events after application of PregS, although the IEI did show a measurable reduction post PregS (Fig. 6b, c).

We confirmed that the reduction in LFP spiking activity was associated with a rescue of the synaptic abnormalities by performing whole-cell patch clamp recordings in hippocampal slices from animals exposed to short (48 h) or more chronic (7 days) NMDAR antibody exposure. In both conditions, PregS increased the frequency of sEPSCS recorded in CA3 neurons from NMDAR antibody treated animals back to control levels (as measured by a fall in IEI; Fig. 7a, b). There was no significant effect of PregS on sEPSC amplitude ($15.2 \pm 0.93$ vs $14.5 \pm 1.3$, $n = 12$; $p = 0.5$, Wilcoxon ranked pairs test), mini-IPSC amplitude ($18.9 \pm 4.1$ vs $15.9 \pm 2.6$ pA, $n = 6$ in both groups; $p = 0.15$, Wilcoxon ranked pairs test) or mini IPSC IEI ($200 \pm 39$ vs $60.2 \pm 69.3$ ms, $n = 6$ in both groups; $p = 0.22$, Wilcoxon ranked pairs test).

Finally, we showed that this acute reversal of NMDAR antibody mediated electrophysiological effects by PregS could in principle be translated to human brains. Paediatric brain slices prepared from tissue resected during epilepsy surgery (Fig. 7c) was exposed to PregS. As in the rat brain slices, we saw a similar increase in the frequency of sEPSCs (i.e. reduction of IEI of sEPSCs) but no effect on the sEPSC amplitude (Fig. 7d, e). There was also no significant change in the sIPSCS amplitude after addition of PregS to human tissue ($31.87 \pm 9.7$ vs $27.96 \pm 4.1$, $n = 6$; $p = 0.44$, Wilcoxon paired rank test), or in the IEI ($123 \pm 41.8$ vs $95.7 \pm 15.8$; $p = 0.09$, Wilcoxon paired rank test).

## Discussion

NMDAR antibodies disrupt normal brain circuit function and are associated with a range of neurological symptoms in patients, including epileptic seizures. The majority of these symptoms are now understood to be directly caused by antibody-induced hypofunction of the excitatory NMDAR. However, the link between reduced NMDAR function and increased seizure propensity has remained unexplained, and is particularly counterintuitive from a classical perspective of seizures arising from an excitatory-inhibitory imbalance[29]. Combining electrophysiological recordings in a rat

model of NMDAR antibody encephalitis and computational modelling of brain microcircuit function, we show that an effective reduction in excitatory synaptic neurotransmission observed at the single-neuron scale underlies NMDAR antibody epileptogenesis by altering the dynamical makeup of brain microcircuitry, making it more susceptible to excursions into pathological brain states. We fitted synaptic parameters of microcircuit models to spectral features of LFP data in different experimental conditions corresponding to a non-epileptic, interictal condition (control), an epileptic and interictal condition (NMDAR antibody interictal) and an epileptic and ictal condition (NMDAR-ictal). This allowed the separation of two main effects—*epileptogenesis*, and *ictogenesis*. In terms of parameter space, these effects were separable, meaning that they acted on different subsets of synaptic parameters. The transition from interictal to ictal state (i.e. ictogenesis) was best explained by changes in synaptic coupling that effectively push excitation-inhibition out of balance into a hyperexcitable state. Yet the pathophysiological process of epileptogenesis—i.e. the synaptic changes that render the brain more likely to undergo an interictal-ictal transition—are separable from those associated with ictogenesis. In fact, in our particular animal model we can show that the *hypo*excitation induced by NMDAR antibody counterintuitively contributes to epileptogenicity, by increasing the sensitivity of microcircuits to fluctuations of the ictogenicity parameters. We demonstrate that this phenotype can be rescued with the neurosteroid, PregS, which, among other pharmacological actions, is known to increase NMDAR availability at the neuronal membrane. Furthermore, we reproduced the synaptic effect of PregS in human brain tissue in vitro, indicating its potential use as precision-treatment for epileptic seizures in patients with established NMDAR antibody encephalitis and potentially other seizure aetiologies associated with NMDAR-hypofunction.

The genetic background of rodent models can confer different levels of seizure susceptibility[30,31]. Juvenile Wistar rats are highly seizure susceptible at P21[32] and proved to be an effective choice for this passive transfer NMDAR antibody-mediated seizure model. The CA3 region was found to have the most spontaneous epileptic activity after NMDAR antibody ICV injection, and epileptic discharges were therefore recorded from this location in vivo. The CA3 hippocampal region is also the seizure-onset zone in other murine models of epilepsy[33,34]. Another study using NMDAR antibody positive CSF recorded a higher incidence of CA3 epileptiform afterpotentials in brain slices from Wistar rats injected with NMDAR antibodies compared to controls[35]; these epileptiform events were not seen in other hippocampal areas[36,37]. Furthermore, electrophysiology studies with leucine-rich-glioma-1 autoantibodies (LGI1 antibodies), that cause facio-brachial dystonic seziures and limbic encephalitis, enhanced CA3 pyramidal cell excitability when applied in vitro[38,39], and patients show focal CA3 subfield atrophy on 7T MRI following LGI1 antibody-mediated disease[40]. Together, these reports and our results confirm that the CA3 subfield is an important epileptogenic hippocampal region in immune-mediated types of epilepsy, justifying its further investigation.

Almost all healthy brains can produce epileptic seizures in the context of acute excitation-inhibition imbalance, such as acute GABA-blockade caused by chemoconvulsants[41]. Furthermore, many effective anti-seizure drugs aim to restore this balance by increasing levels of inhibition or reducing neuronal excitation[42]. Hypofunction in both excitatory AMPAR and NMDA receptors have been reported in epilepsies and seizure models, including antibody-induced loss of AMPARs[43,44]. Similarly, recent work in a rat model of chronic temporal lobe epilepsy (TLE) demonstrates that dysfunctional regulation and resultant hypofunction of GluA (AMPAR) subunits is an important factor in epileptogenesis[45]. Defects in another AMPAR subunit, GluA2, are also found in neurodevelopmental disorders such as epileptic encephalopathy

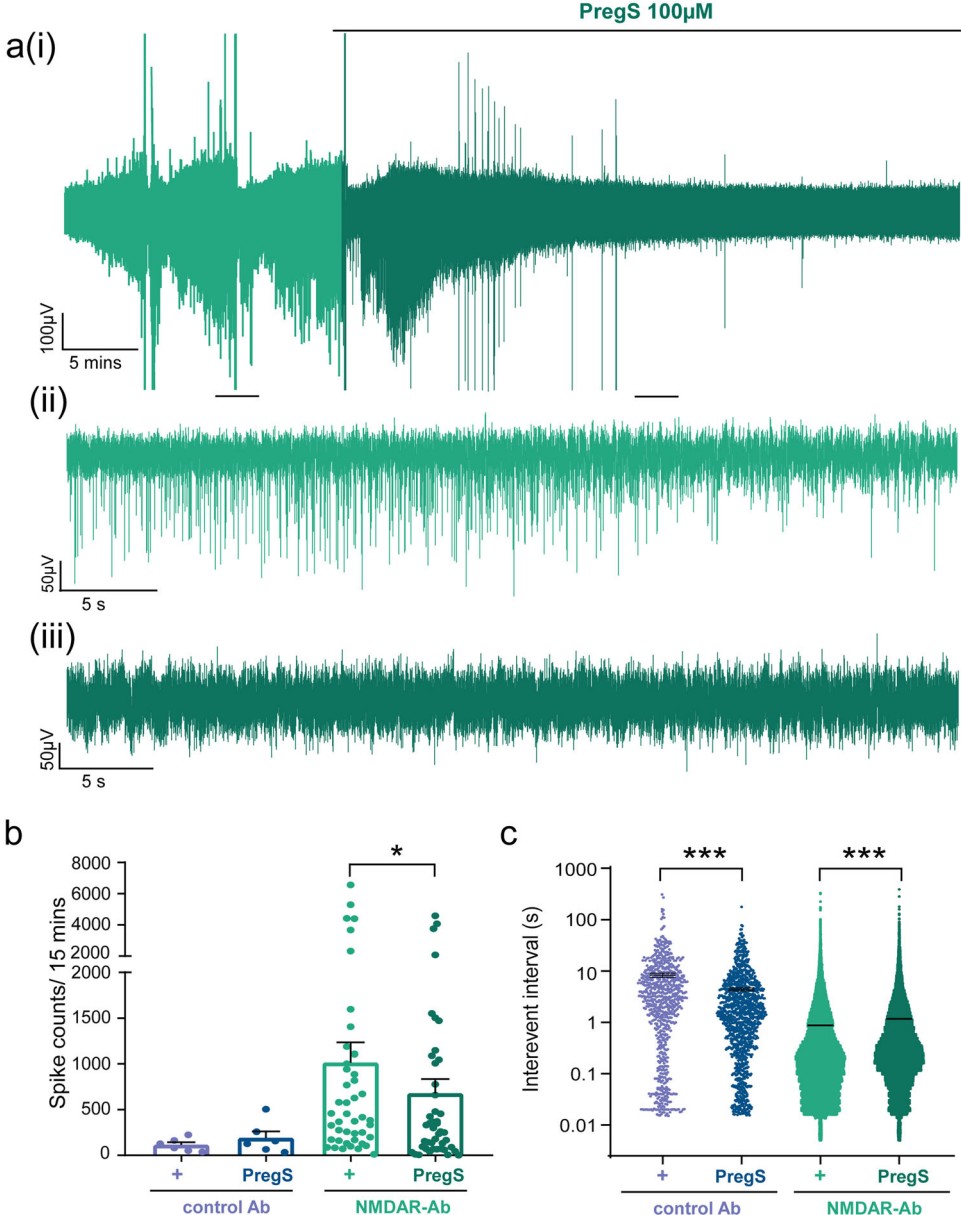

**Fig. 6 Pregnenolone sulphate reduces NMDAR antibody mediated spontaneous ictal activity in vitro. a** Representative in vitro slice LFP recording (subpanel **a**i) from a 7-day NMDAR antibody treated animal illustrates the response to PregS 100 μM. Scale bar 10μV vs. 5 mins. Subpanels **a**(ii) and **a**(iii) represent the magnification of control and PregS 100 μM trace area respectively. Scale bars 50 μV vs. 5 s. **b** Ictal spike counts after PregS is applied to control antibody infused slices (*n* = 6 slices; *p* = 0.07, Wilcoxon paired rank test) and NMDAR antibody infused slices (*n* = 45 slices; *p* < 0.05, Wilcoxon paired rank test). **c** IEI recorded in the control Ab infused LFP slices after addition of PregS indicates an increased frequency of events (*n* = 6 slices; *p* < 0.001, Wilcoxon paired rank test). The frequency of events is reduced in the NMDAR antibody slices after addition of PregS (*n* = 36 slices; ***p* < 0.001; Wilcoxon paired rank test). Measurements expressed as mean (M) ± standard error of the mean (SEM).

in children[46]. Previous experiments have also demonstrated that the weakening of excitatory synapses has a paradoxical effect on neural networks, with one early in vitro and computational study showing use of AMPAR blockers (CNQX) at moderate concentrations evoked seizure-like activity[47] and, more recently, AMPAR antibodies being demonstrated to produce weakened excitatory and inhibitory transimission and increased intrinsic excitability[48] similar to the effects demonstrated here with NMDAR antibodies. In fact, early experiments showed epileptic seizures can be sustained even when measured extracellular calcium reached levels that would completely block chemical synaptic transmission[49] highlighting the role of nonsynaptic neural network properties in epileptogenesis. Finally, as the

Turrigiano laboratory has shown, synaptic scaling of excitatory amino acid receptors is co-regulated alongside intrinsic excitability[48] through the actions of CaMKIV, but GABA inhibition is separately regulated, emphasising the interdependence of synaptic, cellular and circuit mechanisms in control of network excitability[50].

That NMDAR hypofunction is proconvulsant has been shown though a variety of approaches. Genetically modified mice lacking NMDARs in CA3 pyramidal neurons were more susceptible to kainate-induced seizures, and pharmacological blockade of CA3 NMDAR in adult wild-type mice produced similar results[51]. Loss of function mutations in both the GluN1 and GluN2A subunits of the NMDAR have been identified in patients with epilepsy and

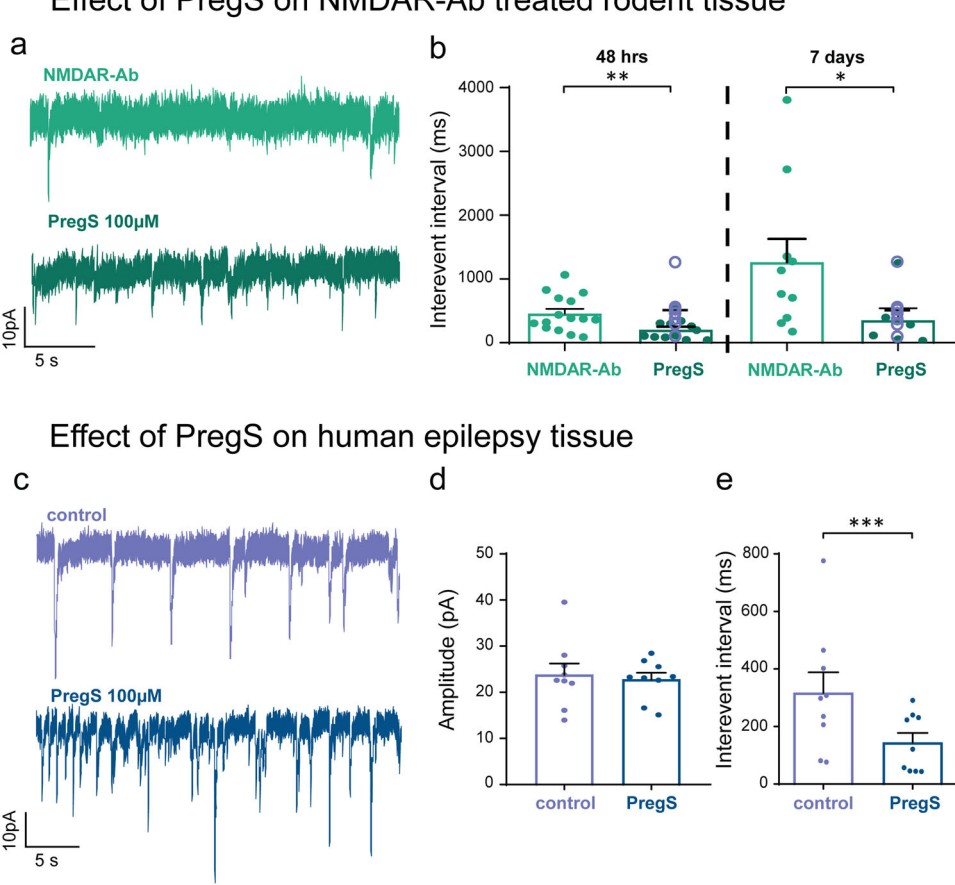

**Fig. 7 Pregnenolone sulphate increases glutamatergic transmission in NMDAR antibody infused rodent hippocampal slices in vitro and has similar effects in human epilepsy brain tissue. a** Representative whole-cell patch clamp recording of putative CA3 pyramidal cell sEPSCs from an NMDAR antibody treated hippocampal slice before and after addition of PregS. Scale bar 10 pA vs. 5 s. **b** IEI of sEPSCS recorded from putative CA3 pyramidal cells before and after PregS application to hippocampal slices prepared from NMDAR antibody treated animals at 48 h after Ab injection ($n = 12$ pairs of cells; **$p = 0.0093$, Wilcoxon paired rank test) and after 7 day NMDAR antibody infusion ($n = 6$ pairs of cells; *$p = 0.03$, Wilcoxon paired rank test). The IEI of sEPSCs recorded from CA3 pyramidal cells in control antibody slices are added for comparison ($n = 7$ cells; shown as open purple circles superimposed on PregS bar). Statistical comparison between the three groups (NMDAR antibody vs. NMDAR antibody + PregS vs. control) was significant at the 48 h ($p = 0.02$, Kruskal–Wallis test) and 7 day timepoints ($p = 0.01$, Kruskal–Wallis test). **c** Representative in vitro whole-cell patch clamp recordings of sEPSCs in human paediatric epilepsy before and after addition of PregS. Scale bar 10 pA vs. 5 s. **d** Amplitudes of sEPSCs recorded from cells in human brain tissue before and after addition of PregS in vitro ($n = 9$ cells; $p = $ ns (1), Wilcoxon ranked pairs test). **e** Interevent interval of sEPSCS recorded from cells in human brain tissue before and after addition of PregS in vitro ($n = 9$ cells; ***$p = 0.004$, Wilcoxon ranked pairs test). Measurements expressed as mean median (M) ± standard error of the mean (SEM).

neurodevelopmental disorders[52,53]. A recent dynamic causal modelling study of NMDAR antibody encephalitis patient EEGs also showed that the deficit in signalling at NMDARs in patients with NMDAR antibody encephalitis was predominantly located at the NMDA receptors of *excitatory* neurons rather than at inhibitory interneurons[54], as shown here in our model. By combining detailed multiscale electrophysiology and computational modelling, we provide further evidence that the direct synaptic effects of NMDAR antibody identified from whole-cell patch clamp recordings offer a quantitative explanation for the epileptic brain dynamics captured in vivo using local field potentials.

In our circuit models of NMDAR antibody-induced epileptogenesis, the parameter changes required to optimally model the associated LFP changes required not only changes in NMDAR-transmission related parameters, but also more widespread changes involving both excitatory and inhibitory neuronal populations. This would appear to fit with the co-dependent changes seen in our rodent model and those of others as reported

above. In addition to such compensatory and maladaptive network changes, heterogeneity of NMDAR distribution at normal baseline function may also account for the unexpected effect of NMDAR-hypofunction in neuronal circuit dynamics. Selective forebrain pyramidal neuron NR1 knockout mice demonstrated both socio-cognitive deficits and *increased* pyramidal cell excitability[55], potentially driven by a release from inhibition by reducing putative NMDAR-mediated excitatory driving of inhibitory neuron activity, or by changing phase relationships between interacting neuronal populations. Similarly, hippocampal CA3 NMDA receptors may normally act to suppress the excitability of the recurrent network by restricting synchronous firing of CA3 neurons[51], with NMDAR-blockade releasing them from this suppression. Our modelling also suggests that NMDAR antibody hypofunction may have stronger impacts on specific compartments of the affected microcircuitry.

Lastly, the integrated brain (micro-)circuits under investigation here show complex emergent dynamics arising from a range of often nonlinear interactions. This complexity can often give rise

to unpredictable results, even where simple systems show a clear mapping between excitation/inhibition balance and seizure propensity. Our modelling suggests that the effective synaptic connectivity changes associated with NMDAR antibodies sensitise the circuit to perturbations in another (largely unrelated) set of parameters. This insight implies that whilst the seizure transition may be indicative of a change in a balance of parameters associated with excitation and inhibition, there may be territories of parameter space where this change is tolerated without any notable impact on the local field potential and other territories where the same change of excitation/inhibition parameter balance would produce large changes consistent with an epileptic seizure. The simulations here indicate that the NMDAR antibody models are much more sensitive to 'ictogenesis'-associated parameter changes, suggesting that the combined parameter changes induced by NMDAR antibody have pushed the microcircuit into a territory in parameter space, where it is particularly sensitive to further perturbations.

Neurosteroids are derived from cholesterol and are precursors to gonadal steroid hormones and the adrenal corticosteroids. They are produced de novo by glial cells and principal neurons, hence the term 'neurosteroid'[56], and modulate brain excitability primarily by interacting with receptors and ion channels on the membrane surface rather than within cells[57]. Pregnenolone (precursor to PregS) elevates levels of allopregnanolone and pregnenolone sulphate, a positive NMDAR receptor modulator. Pregnenolone has been used as an effective adjunctive therapy in treatment trials for schizophrenia, a psychiatric condition that shares the pathophysiology of NMDAR hypofunction[58–60]. These trials demonstrate both the safety and efficacy of Pregnenolone in humans. In vivo, pregnenolone was found to reduce the hyper-locomotion, stereotypic bouts and prepulse inhibition deficits in a mouse model of schizophrenia acutely, and improved impaired cognitive deficits with chronic treatment[61]. PregS, a sulfated ester of pregnenolone, is a positive NMDAR modulator[62,63] and studies show that the mechanism of action is to increase surface expression of functional NMDARs containing the NMDAR subunit subtype 2A (NR2A) or subtype 2B (NR2B), both the most abundant subunits found in the brain; no evidence points to changes in NR2C or NR2D subunits[28]. In a previous study, this membrane delimited effect postulated to be via a non-canonical, G-protein and calcium-dependent mechanism occurred within 10 min of application to cortical neurons, increasing surface levels by 60–100%[28]; this speed of action of PregS was reproducible in vitro in our brain slice recordings. In addition to the effect of PregS on NMDARs, other CNS targets have been described, including the GABA$_A$ receptor, glycine receptor, and more recently the TRPM3 channel[64] and Kir2.3 channel[65]. The effects of PregS on the GABA$_A$R most commonly described are at lower, nanomolar range doses[66,67]. We found that the 100 µM dose of PregS had no effect on GABAergic neurotransmission in the brain slices from the NMDAR antibody model. A proconvulsant effect was seen in the control antibody slices, but the reverse was seen in the NMDAR antibody infused brain slices, suggesting that it was the increase of NMDARs that led to the reduction in the frequency of epileptic events.

Previous studies with NMDAR modulators and other treatments have focused on the prevention of pathogenic antibody effects[9–11]. To our knowledge this is the first study to show that symptoms established for more than 48 h (in this case ictal activity in vitro), can be treated with the use of the neurosteroid PregS. Whilst PregS was chosen here specifically to treat the known underlying pathophysiology, its role for the treatment of epilepsy may extend into other conditions or scenarios where the treatments that accord to the notion of reestablishing excitation-inhibition balance have failed to be effective (e.g. refractory status epilepticus)[68].

We did not record directly from inhibitory interneurons but it is likely that both cell types are involved in driving the epileptogenesis. The CA3 network is highly complex, with CA3 pyramidal cells having convergent and divergent connections with several interneuronal types in different hippocampal regions[69]. A single CA3 pyramidal cell can make synapses with 30,000–60,000 neurons in the ipsilateral hippocampus[70]. It, therefore, remains a possibility that NMDAR hypofunction affecting GABAergic interneuronal populations produces a disinhibited CA3 network[55,71,72] and investigating this possibility should be a goal of further electrophysiological studies.

Another limitation to this study was that the cognitive effects of NMDAR antibodies were not investigated in the animals or electrophysiologically through long-term potentiation (LTP) recordings. However, LTP recordings will be difficult to interpret with spontaneous ictal spikes as seen in these brain slices. It is known that in CA3, NMDAR-dependent (associational-commissural, A-C fibres) and NMDAR-independent (Mossy-fibre) forms of LTP are expressed in adjacent synapses. LTP has been shown to be reduced by NMDAR antibody positive CSF (experiments performed in Wistar rats), and this reduction may contribute to the cognitive deficits seen clinically[35]. Mossy fibre synapses display a markedly lower proportion of GluN2B-containing NMDARs[73] than associative or commissural synapses, therefore a loss of NMDARs in this critical population of cells could also be devastating. PregS offers an attractive treatment strategy to increase NR1 and NR2B subunit NMDARs[28] that could potentially mitigate the short and long-term cognitive effects seen in patients, as well as the seizures. Further detailed molecular studies will need to be done to characterize the NMDAR and subunit changes.

Future studies also need to focus on ascertaining the optimal brain concentrations of the neurosteroid PregS in vivo in the NMDAR antibody mediated disease models, and on determining whether these can be achieved through direct administration of PregS (e.g. by subcutaneous injection) or the precursor Pregnenolone (oral administration, a treatment already used safely in humans)[60]. There is precedence for the use of neurosteroids in other forms of epilepsy, including genetic epilepsies (e.g. Phase 3 clinical trial of ganaxolone in *PCDH-19* female patients; NCT03865732) as well as refractory status epilepticus[74]. In genetic epilepsy, research efforts have focused for some time on modifying the specific underlying genetic effect with new and repurposed drugs[75]. In this study, we show that there is also a possibility of providing receptor-specific neurosteroid treatment to affected patents with immune-mediated epilepsy and encephalitis.

## Methods

**NMDAR antibody and control antibodies**. Three human-derived NMDAR antibody preparations were obtained for experimental use. The first was NMDAR antibody positive IgG prepared from plasma obtained, with informed consent at plasmapheresis from a patient (11-year-old female with NMDAR antibody encephalitis), using Protein G beads[18]. In brief, plasma was diluted 1:1 with Hartmann's solution and passed through Protein G sepharose column beads. To elute IgG, 0.1 M glycine solution (pH 2.3) was used and the sample then neutralized with 1 M Tris pH 8. A Coomassie Plus assay kit (Pierce, USA) was used to determine protein concentrations of eluted fractions. Peak protein fractions were pooled and dialysed against Hartmann's solution, and IgG concentrated using Amicon ultra-4 10KDa filter units, and then filter-sterilized. Nanodrop 3300 (ThermoScientific, UK) was used to determine final IgG concentration (37 mg/ml). In addition, two different recombinant NMDAR antibody specific human monoclonal antibodies were donated for use; SSM5 (4.3 mg/ml)[8] and 003-102 (3.4 mg/ml)[3]. Both of these NMDAR antibody monoclonal antibodies have previously been shown to have in vitro and in vivo pathogenic effects[3,8].

Control antibodies (Control antibodies) were human derived IgG from one healthy individual (healthy male, aged 35 years; concentration 30 mg/ml), and two non-brain reactive monoclonal Control antibodies; mGO53 (514.5 µg/ml)[3], and 12D7 (6 mg/ml)[8].

As all antibody preparations had previously been used in the studies specified and were known to be pathogenic, and stocks for some preparations were limited, a direct comparison was not made between each antibody preparation of the epileptogenic effects. The three NMDAR antibody preparations will be collectively referred to as "NMDAR antibodies" throughout the paper and the three control preparations as "Control antibodies". Detailed description of antibody use per experiment is provided in Supplementary Table 1.

**Animals.** Given our experience in other rodent epilepsy models[32,45], we hypothesized that the use of juvenile male Wistar rats would increase the probability of observing spontaneous seizures in a NMDAR antibody passive transfer model. Thirty-one post-natal day 21 (P21) male Wistar rats, weighing 50–58 g, were used for the in vitro experiments and 12 were used for the in vivo experiments. The animals were housed in temperature- and humidity-controlled conditions with a 12 h/12 h light/dark cycle, and allowed to feed and drink ad libitum. All procedures were compliant with current UK Home Office guidelines as required by the Home Office Animals (Scientific Procedures) Act 1986. They were carried out under the authority and procedural approval of a UK Home Office approved project license and in line with ARRIVE guidelines. Local ethical approval for the study was granted by the Aston Bioethics Committee, University of Aston, Birmingham, UK.

**Surgery: stereotaxic ICV injection of NMDAR- and control antibodies.** ICV injection was performed under isoflurane anaesthesia with rats fixed in a stereotaxic frame[18]. A 0.5 cm incision was made caudally over the skull surface which was then exposed before a single burr hole was made for ICV injection. The co-ordinates for the left lateral ventricle were 1.5 mm left lateral and 0.6 mm caudal from bregma[76]. 10 micrograms (10 mcg) of SSM5 (NMDAR antibody monoclonal antibody), or non-reactive control antibody (12D7) or 8 µL of NMDAR antibody IgG[18] or healthy control IgG was used. Antibody preparations were coded to allow for blinded analysis. The antibody was injected over twenty minutes using a Hamilton syringe with 33 gauge needle to a depth of 3.2 mm from skull surface. Following injection, the syringe was left in place for ten minutes before being withdrawn over twenty minutes. The skin was sutured and animal allowed to recover before being replaced in home cage.

**Local field potential (LFP) recordings.** Forty-eight hours after ICV injection or at the end of the recording period in telemetry experiments, rats were anaesthetized using isoflurane followed, after loss of consciousness, by pentobarbital (60 mg/kg, SC) and xylazine (10 mg/kg, IM). Transcardial perfusion was then performed using ice-cold artificial cerebrospinal fluid (cutting aCSF) in which NaCl had been replaced with equimolar sucrose. Animals were decapitated, and the brain was removed and placed in ice-cold cutting aCSF. Sagittal slices at 450µm were cut using a vibratome (Campden Instruments, UK)[77]. The slices were transferred to a storage chamber containing standard aCSF (in mM: 125 NaCl, 3KCl, 1.6 MgSO₄, 1.25 NaH₂PO₄, 26 NaHCO₃, 2 CaCl₂, 10 glucose) and constantly bubbled with carbogen (95% O₂/5% CO₂) at room-temperature for at least one hour prior to recording. Slices were then transferred to an interface chamber (Scientific System Design Inc., Canada) and continuously perfused with aCSF (2–3 ml/min) maintained at 30–31 °C and visualised with a SZ51 stereomicroscope (Olympus). A Flaming-Brown micropipette puller (Sutter Instruments, CA) was used to pull borosilicate glass microelectrodes with an open tip resistance of 1–3 MΩ when filled with aCSF. Electrodes were inserted into CA3 and CA1 regions of the hippocampus using Narishige MM-3 (Narishige, Japan) micromanipulators and signals were recorded using an EXT-02 headstage and amplifier (NPI Electronic). HumBugs (Quest Scientific) were used to remove electrical 50 Hz interference from recordings. The signal was then amplified ×100 via an EX10-2F amplifier (NPI Electronics) and filtered (700 Hz low-pass, 0.3 Hz high-pass, sampling rate 5 kHz) and further amplified ×10 using a LHBF-48X amplifier (NPI Electronics). Signals were then digitised at 10 KHz using an analogue to digital converter (Micro-1401-II; CED). LFP recordings were assessed using Spike2 software (CED) for spontaneous epileptiform activity. Spike2 was used to calculate the root mean square (RMS) amplitude of each recording. Epileptiform activity was classified as an event when it displayed an amplitude greater than four-fold the RMS amplitude, providing the event count, while the time difference between these events provided the interevent interval. A custom MATLAB script was used to prevent false event detection, removing those with an interevent interval less than 15 ms. Statistical analysis was conducted using Graphpad Prism 8 (GraphPad Software Inc), statistical tests used to compare groups (Mann-Whitney) were one-tailed. Measurements were expressed as mean (M) ± standard error of the mean (SEM).

**Whole-cell patch clamp recordings.** For whole-cell patch clamp recordings, brain slices were prepared as above but sliced at a thickness of 350 µm. For recording, the slices were transferred to a submerged chamber (Scientifica, UK) and visualised using Nomarski infra-red optics on a BX51WI Microscope (Olympus). A Flaming-Brown micropipette puller (Sutter Instruments, USA) was used to pull borosilicate glass microelectrodes of 4–6 MΩ for recordings. For voltage clamp recordings, electrodes were filled with an internal solution containing (in mM): 100 CsCl, 40 HEPES, 1 Qx-314, 0.6 EGTA, 5 MgCl₂, 10 TEA-Cl, 4 ATP-Na, 0.3 GTP-Na and 1 IEM 1460 (titrated with CsOH to pH 7.25) at 290–305 mOsm for IPSCs. For EPSCs

the internal solution contained (in mM): 40 HEPES, 1 Qx-314, 0.6 EGTA, 2 NaCl, 5 Mg-gluconate, 5 TEA-Cl, 10 Phospho-Creatinine, 4 ATP-Na, 0.3 GTP-Na (titrated with CsOH to pH 7.3) at 285 mOsm for EPSCs. The EPSCs and IPSCs were recorded as apparent inward currents at −70 mV using an Axopatch 200B amplifier (Molecular Devices, USA). Signals were low-pass filtered at 5 kHz with an 8-pole Bessel filter and digitised at 10 kHz using a Digidata 1440 A and pClamp software (Molecular Devices). For intrinsic cell excitability experiments, whole-cell current-clamp recordings were made from hippocampal neurons using an Axopatch 700 series amplifier (Molecular Devices, USA) and with a K-gluconate-based intracellular solution containing (in mM) K-gluconate (120) hepes (40) KCL (10) Na2ATP (2) Na-GTP (0.3) MgCl₂ (3) EGTA (0.5). Membrane potential was manually adjusted when required to −60 mV by continuous somatic current injection. Membrane response and action potential firing were measured by applying depolarizing steps of current (duration of 1 ms at 10pA increments). For spike/current intensity curves, the number of action potentials was plotted against the injected current. Input resistance was measured from the slope of a linear regression fit to the voltage–current plotted graph between −80 pA to 0 pA. Data were analysed using Axograph and Prism 8, statistical tests used to compare groups (Mann–Whitney and Wilcoxon matched pairs ranked tests) were one-tailed. Measurements are expressed as mean median ± SEM.

For human tissue recordings, brain slices were prepared as above using choline-based artificial cerebrospinal fluid[78,79]. Briefly, human tissue was obtained with informed parental or guardian consent from paediatric patients prior to them undergoing epilepsy surgery at Birmingham Children's Hospital. Ethical approval was obtained from the Black Country Local Ethics Committee (10/H1202/23; 30 April 2010), and from Aston University's ethics committee (Project 308 cellular studies in epilepsy) and through the Research and Development Department at Birmingham Children's Hospital (IRAS ID12287). Brain tissue was obtained from 7 patients (F:M 4:3), median age 6 years (range 4–16 years). Surgical procedures included frontal resection (three), temporal parietal occipital disconnection (one), temporal lobectomy (two), and right hemispherectomy (one) as described in Supplementary Table 2. Specimens were resected intraoperatively with minimal traumatic tissue damage, and minimal use of electrocautery. Brain tissue was transported in ice-cold choline-based artificial CSF as previously described[79].

**Drugs.** For all in vitro electrophysiology experiments, Pregnenolone sulphate (Sigma, UK) was prepared as 1 M stock using dimethyl sulfoxide (DMSO). A concentration of 100 µM was used for all electrophysiology experiments. MK-801 (Hello Bio, UK) was prepared with distilled water as 10 mM stock and used at 10 µM concentrations.

**Immunofluorescence and image analysis.** Following in vitro electrophysiology experiments, brain slices were briefly fixed in 4% paraformaldehyde (PFA) for 45 min. To determine NMDAR antibody or control antibody binding, the sections were rinsed with phosphate-buffered saline (PBS) and then incubated in anti-human IgG Alexa-Fluor-488 at 1 in 1000 (Invitrogen, UK) overnight at 4 °C. Sections were washed and mounted with aqueous mounting medium containing DAPI. Images of hippocampal sections were taken on a Tandem Confocal Scanning SP5 II microscope (Leica Microsystems Ltd) using a (10×/0.30) dry objective lens. Fluorescence was excited with a 488 nm argon laser at 23% power (emission bandwidth 504nm-564nm). Confocal micrographs were acquired at 1024 pixels2 with actual area of 1.48mm2, and scanning speed 100 Hz.

Biocytin (Sigma, UK) was added to the electrode filling solution at a concentration of 5 mg/ml to determine the characteristics of the recorded cell during whole-cell patch clamp experiments. Neurons were filled with biocytin and slices were fixed overnight at 4 °C in 100 mM PBS (pH 7.3) containing 4% PFA (BDH, USA). After washing in PBS, slices were incubated for 18 h in PBS supplemented with 1% Triton X-100 and 0.2% streptavidin Alexa Fluor 488 conjugate (Invitrogen) at 4–6 °C. Sections were washed and mounted with aqueous mounting medium containing DAPI for imaging.

Fluorescent intensity Log $EC_{50}$ ratio of hippocampal sections was determined using 5 regions of interest (ROI) of the same size from the molecular outer layer and from the granular cell layer of CA3. The mean fluorescent intensity Log $EC_{50}$ value of the outer molecular layer was normalised by dividing it by the mean fluorescent intensity Log $EC_{50}$ of nonspecific staining in the granular cell layer. This process provided a ratio for each replicate. All images were analysed through FIJI by converting them to grayscale (8-bit) before generating cumulative pixel intensity histograms for each ROI which were calculated using a customized macro programme[1,18]. Any statistical analysis performed was conducted in GraphPad Prism 8.

**Surgery: placement of ventricular catheters, osmotic pumps and wireless EEG transmitters.** Osmotic pumps (model 1007D, Azlet) were used for cerebroventricular infusion of monoclonal antibodies and human IgG (volume 100 µl, flow rate 0.5 µl, duration 7 days)[7,8]. Two osmotic pumps per animal were prepared by loading with monoclonal antibodies or human IgG and then connected to polyethylene tubing 69 mm-×-1.14 mm diameter (C312VT; PlasticsOne) and a double osmotic pump connector intraventricular cannula (328OPD-3.0/SPC; PlasticsOne). Pumps were left overnight in sterile saline solution at 37 °C. Rats were placed in a

**Table 1 Free parameters fitted by the DCM (parameters related to NMDAR-transmission in bold).**

| | |
|---|---|
| $\tau_1$ (spiny stellate time constant) | $\boldsymbol{\gamma_1}$ (**ss to ii excitation**) |
| $\boldsymbol{\tau_2}$ (**inhibitory interneuron time constant**) | $\boldsymbol{\gamma_2}$ (**dp to ii excitation**) |
| $\boldsymbol{\tau_3}$ (**superficial pyramidal cell time constant**) | $\boldsymbol{\gamma_3}$ (**ss to sp excitation**) |
| $\tau_4$ (deep pyramidal cell time constant) | $\gamma_4$ (sp to ss inhibition) |
| $\boldsymbol{\sigma}$ (**population response variance**) | $\gamma_5$ (ii to ss inhibition) |
| | $\gamma_6$ (ii to ip inhibition) |
| | $\gamma_7$ (sp inhibitory self-modulation) |
| | $\gamma_8$ (ii inhibitory self-modulation) |
| | $\gamma_9$ (ss inhibitory self-modulation) |
| | $\gamma_{10}$ (dp inhibitory self-modulation) |

stereotaxic frame for surgery performed under isoflurane anaesthesia. Osmotic pumps were placed subcutaneously and attached cannulae inserted into both lateral ventricles (1.5 mm lateral, 0.6 mm caudal). The EEG transmitter was placed in a subcutaneous pocket formed over the right flank with a single skin incision and blunt tissue dissection (A3028B-DD subcutaneous transmitters, 90-mm leads, OpenSource Instruments (OSI)), and depth electrode (W-Electrode (SCE-W), OSI) placed in the left hippocampus (CA3, 3.5 mm lateral, 3.6 mm caudal, depth 2.3 mm) with a reference electrode implanted on the contralateral skull surface (3.5 mm lateral, 3.6 mm caudal). The cannula and skull electrodes were secured with dental cement[18,25].

**Collection and analysis of EEG data**. EEG data (wide band pass 0.2–160 Hz sampled at 512 samples per second) was collected and recorded from freely moving animals placed in a custom-built Faraday cage with an aerial. Transmitter signals were continuously recorded using Neuroarchiver software (OSI) and analysed as previously described[18,25,80]. In brief, for automated ictal event detection, video-EEG matching was used to identify ictal EEG events. The Event Classifier (OSI) was then used to classify one-second segments of EEG according to program metrics (power, coastline, intermittency, coherence, asymmetry, spikiness) enabling similar events to cluster together when plotted. This generated a library of ictal events that allowed fast identification of abnormal EEG events by automated comparison to the library (http://www.opensourceinstruments.com/Electronics/A3018/Seizure_Detection.html). Powerband analysis was carried out using a custom-designed macro. Statistical analysis was conducted using Graphpad Prism 8 (GraphPad Software Inc.).

**Statistics and Reproducibility**. Data in each graph was analysed with two-way ANOVA and post-hoc test, Mann–Whitney or Wilcoxon paired rank test, specified in figure legend. In in vitro electrophysiology experiments, individual points represent brain slices or cells recorded from at least three different animals or human brain tissue samples per experimental group. In in vivo experiments, data were collected from six animals in each group. The reproducibility of the data were maintained by using different NMDAR-Ab preparations. All graphs are expressed a mean (M) with bars representing standard error of the mean (SEM).

**Dynamic causal modelling**. Dynamic causal modelling (DCM) was performed using the academic software package SPM12 (https://www.fil.ion.ucl.ac.uk/spm/). All analysis code and raw data are available online at https://doi.org/10.17605/OSF.IO/GUPBF[81]. Analogous to previous work[21,82], we modelled LFP spectra recorded in rat hippocampal tissue as arising from a single canonical microcircuit at steady state[83]. This 'canonical microcircuit' is a layered set of recurrently coupled excitatory and inhibitory neural masses[84–86]. Originally conceived as a description of layered cortical population activity, the canonical microcircuit model has also been shown to offer a parsimonious explanation of rodent hippocampal activity in different physiological and abnormal states[87]. The four populations of the canonical microcircuit act as two coupled oscillator pairs: One superficial oscillator typically characterised by fast (beta/gamma-range) frequencies and consisting of 'superficial pyramidal cells' and 'spiny stellate cells'; and one deep oscillator with typically slow (theta/alpha-range) frequencies consisting of inhibitory interneurons and deep pyramidal cells. This model is a description of neuronal interactions at the population level. Effective synaptic connectivity is parameterized by three key parameters: effective coupling strength ($\gamma$), a postsynaptic time constant ($\tau$), and the slope of a parameterised sigmoid function ($\sigma$) that encodes population response variance[88]. The free neuronal parameters in the model implemented here are summarized in Table 1.

DCM employs a standard variational Laplace model inversion to (i) estimate posterior densities of model parameters given the data and some prior values, and (ii) provide an estimate for the Bayesian model evidence using the free energy approximation[89,90]. Initial parameter values were taken from the standard formulation of the canonical microcircuit in dynamic causal modelling originally derived from cat visual cortex recordings[27,84]. These were adapted to this particular experiment in a two-stage approach—posterior expected values from an initial model fit to the baseline data were subsequently used as priors in the subsequently fitted conditions. These priors do not define absolute limits to the parameter ranges but restrict excessive between-condition differences through Bayesian integration of prior estimates ('the priors') and empirical likelihoods ('the data') to generate posterior densities ('parameter estimates')[91].

The estimation of the Bayesian model evidence allowed us to compare evidence for models of the same data under different assumptions encoded in different priors. Specifically, we wished to test the hypothesis that NMDAR antibody-induced changes measured using whole-cell patch clamp recordings contribute to the LFP features observed in vivo and that were modelled with the DCM. To test this hypothesis, we, therefore, performed several DCM inversions on the same data under quantitatively different priors, and compared the model evidence for each of these models.

Specifically we followed the following procedure; we first wanted to identify which synaptic parameters we would expect to be altered in the NMDAR antibody in vivo experiments, based on the microscale in vitro recordings. To be able to make quantitative predictions about such parameters we quantified relevant features of synaptic transmission in whole-cell patch clamp recordings. Using these features, we generated quantitative predictions of synaptic parameter differences between control antibody and NMDAR antibody conditions. To be able to infer synaptic coupling parameters underlying the local field potential data, we extracted data from different relevant conditions and states (i.e. interictal control antibody, interictal NMDAR antibody, and ictal NMDAR antibody) to then fit the microcircuit models using DCM to the average power spectral densities of each of these conditions. We next wished to identify a synaptic parameter combination that best describes a physiological, interictal state from which all other conditions emerge by changes in synaptic coupling. We, therefore, fitted a single canonical microcircuit model to control antibody data, using the inferred parameters as priors in subsequent Bayesian model inversions in the next steps. To address the question whether the observed microscale changes in synaptic function contribute to the observed LFP features, we fitted a single canonical microcircuit to interictal NMDAR antibody data under different sets of priors—including those that are informed by microscale data, and those that are not. This allowed us to perform Bayesian model comparison between these models. Here we would expect models that contain microscale-derived prior information to outperform other models if the modelled microscale features contribute to the LFP data. To address the question which synaptic mechanisms best explain condition- and state-differences, we fitted a single canonical microcircuit to seizure NMDAR antibody data, then quantified the parameter differences between different DCM models. The objective here was to quantify the main differences between conditions (control antibody vs NMDAR antibody) and states (interictal vs. seizure) in terms of neuronal population model parameters.

**Reporting summary**. Further information on research design is available in the Nature Research Reporting Summary linked to this article.

## Data availability
The supportive data for this study are available within the manuscript and the Supplementary Information. All source data underlying figures are presented in Supplementary Data 1. All dynamic causal modelling raw data are available online at http://doi.org/2010.17605/OSF.IO/GUPBF. Requests should be made to the corresponding authors for materials, dependent on stock availability.

## Code availability
Code for data analysis for dynamic causal modelling (Figs. 4, 5) can be found here http://doi.org/2010.17605/OSF.IO/GUPBF.

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

## Acknowledgements

We thank Rob Wykes, Jonathan Cornford and Andrea Lieb (University College London, UK) for use of EEG powerband analysis program. We gratefully acknowledge the Children's Epilepsy Surgery Service at the Birmingham Women's and Children's NHS Hospital Trust for human tissue samples as well as the patients and families contributing to the study, and Aston Biomedical Services for laboratory animal care. S.K.W. was funded by an Epilepsy Research UK Fellowship (F3001) and Wellcome Trust Clinical Research Career Development Fellowship (216613/Z/19/Z) during this work. H.P. received support from the German Research Foundation (DFG; grant numbers PR1274/3-1, 4-1, 5-1) and the German Federal Ministry of Education and Research (BMBF; Connect-Generate). N.G. received support from the German Ministry for Education and Research (BMBF; 31P7398 and Connect-Generate), the National Multiple Sclerosis Society (NMSS; BRAVEinMS), the Wellcome Trust (208938/Z/17/Z) and the Forschungskommission of the Medical Faculty of the Heinrich-Heine-University. S.D.G. and D.D. funded in part by the Academy of Medical Sciences (SBF004\1053).

## Author contributions

S.K.W. designed the study, carried out, analysed and interpreted data from all in vivo experiments, the in vitro whole-cell patch clamp experiments, and the in vivo EEG experiments. S.K.W. performed the immunofluorescence experiments. S.K.W. drafted the manuscript. R.E.R. helped design the study, analysed and interpreted all the dynamic causal modelling experiments, and drafted the manuscript. M.A.W. acquired, analysed, and interpreted the in vitro LFP experimental data and drafted the manuscript. M.A.U. analysed and interpreted the in vitro LFP experimental data and videos of experimental subjects, and also drafted the manuscript. D.R.D. and C.C.B. acquired, analysed and interpreted the immunofluorescence data. Additionally, D.R.D. helped draft the manuscript. T.T.W. analysed and interpreted the videos of experimental subjects and helped draft the manuscript. S.B. and N.G. produced, verified and provided NMDAR antibody and control antibody preparations for experimental use. J.K. and H.P. produced, verified and provided NMDAR antibody and control antibody preparations for experimental use. L.J. produced and verified NMDAR antibody and control immunoglobulins for experimental use. D.S.B. supervised the dynamic causal modelling experiments and drafted the manuscript. A.V. designed the study, supervised S.K.W. and drafted the manuscript. S.D.G. supervised S.K.W. with in vivo experiments, analysed and interpreted data from in vivo EEG and in vitro electrophysiology experiments. G.L.W. designed the study, provided supervision and assistance to S.K.W. with in vitro experiments, analysed and interpreted data from in vitro electrophysiology experiments, and drafted the manuscript. All authors substantively revised and approved the submitted version.

## Competing interests

The authors declare no competing interests.
