## [Transparent Peer Review File · Communications Biology]

Reviewers' comments:

Reviewer #1 (Remarks to the Author):

The paper "In vitro characterization and neurosteroid treatment ..." by Wright et al., describes an impressive, combined in vitro, in vivo and in silico study of effects of NMDA receptor antibody (NMDAR-Ab) on neural activity, with a focus on seizures. The findings that reduction of NMDARs leads to increased excitability and seizure activity is counterintuitive as well as interesting. The investigators developed a rat model for studying the NMDAR-Ab effects and include a pharmacological in vitro approach to mitigate the epileptogenic effect of the NMDAR-Ab.

I find that the experimental work is well described and I have mainly minor comments on this component. The only major comment I have for this part of the manuscript is that I don't see an overarching statistical analysis with respect to the NMDAR-Ab, Control-Ab, and Neurosteroid results. It is disappointing that the authors only analyze their experiments separately although they do apply Bonferroni correction in a few cases.

The description of modeling component is difficult to follow and does not allow me to fully assess its value. For instance, the model results are based on time and frequency domain properties of the simulated data, but sample traces and spectra of these are not shown. In a combined approach used here, I would expect to see recorded data (such as depicted in the previous figures) and simulation results side-by-side. A direct comparison and similarity between recorded and simulated data would be convincing since Friston's procedure to select the best amongst different simulations is very useful but won't guarantee that the best result is appropriate. Furthermore, two principal components (Fig. 4e), using PCA, are interpreted as epileptogenesis and ictogenesis axes. The basis for this interpretation is unclear, including the results that build on this distinction (Fig. 5 and associated text) and the first paragraph in the Discussion section.

Details

- The title does not reflect the combined in vitro, in vivo and in silico study.
- What is the oscillation in the 2nd trace in Fig. 3a?
- P.10, 2nd paragraph, line 5 and p.31: How were the domain borders for the parameters in Table 1 established?
- Fig. 4a only shows the effect on excitatory cells and omits the effect on the inhibitory ones.
- I find that the Fig. 4b title 'Microcircuit model' and the diagram are misleading because this presentation suggests that a motif of individual neurons is modeled, while the model is a neural mass model
- The procedure used to get to Fig. 4c,d is hard to follow.
- P.29: What filter settings and sample rate were used for EEG acquisition?
- Discussion: Paradoxical effects of reduced synaptic activity, including excitatory synapses, have been reported previously (e.g., van Drongelen et al., IEEE Trans Neural Syst Rehabil Eng. 2005 Jun;13:236; Pumain et al., Exp Neurol. 1985 89:250; ...). It would be useful to reference and briefly discuss these.

Reviewer #2 (Remarks to the Author):

In this paper, the authors generated an NMDAR antibody induced model of encephalitis in rat and characterized seizure activities in vitro and in vivo. They further applied electrophysiological recording to show a reduced excitatory synaptic activity but no change in inhibitory activity, together with epileptiform activities in field potential recording. Modeling study suggests alterations in dynamic behavior of neural circuits may provide a mechanistic explanation. Interestingly, treatment with

pregnenolone sulphate in vitro in both rodent and human slices resulted in a reduction of ictal activity, providing a potential treatment strategy through increasing NMDA receptors. While the characterization of the model is not new, the work does provide mechanistic insight about this disease and potential novel treatment methods. The study was well-designed and conclusions are supported, however, a few issues need to be addressed.

One major issue is that the authors did not assess and present data on the excitability and intrinsic properties of the recorded CA3 pyramidal neurons (e.g. potential changes in injected current-response curve, resting membrane potential, input resistance, etc). It is possible that hypoexcitability of the excitatory synaptic transmission may cause compensatory changes in neuronal intrinsic properties and excitability, making the neurons hyperexcitable. The paper does not have a direct measurement of related parameters. Without data for neuronal excitability and intrinsic properties, it is difficult to conclude that the network is hypoexcitable and not possible to know the contribution of neuronal intrinsic excitability.

Minor issue: The title of the paper may be modified to include electrophysiological findings, it seems interesting and novel.

Reviewers' comments:

Reviewer #1 (Remarks to the Author):

The paper “In vitro characterization and neurosteroid treatment ...” by Wright et al., describes an impressive, combined in vitro, in vivo and in silico study of effects of NMDA receptor antibody (NMDAR-Ab) on neural activity, with a focus on seizures. The findings that reduction of NMDARs leads to increased excitability and seizure activity is counterintuitive as well as interesting. The investigators developed a rat model for studying the NMDAR-Ab effects and include a pharmacological in vitro approach to mitigate the epileptogenic effect of the NMDAR-Ab.

Reviewer 1 comment 1

I find that the experimental work is well described and I have mainly minor comments on this component. The only major comment I have for this part of the manuscript is that I don't see an overarching statistical analysis with respect to the NMDAR-Ab, Control-Ab, and Neurosteroid results. It is disappointing that the authors only analyze their experiments separately although they do apply Bonferroni correction in a few cases.

Reply

We thank the Reviewer for their positive evaluation of the manuscript.

Overarching statistical analysis

This further statistical analysis with respect to the NMDAR-Ab, Control-Ab and Neurosteroid results has now been done and presented in Figure 7, page 46 lines 9-11:

Statistical comparison between the three groups (NMDAR-Ab vs. NMDAR-Ab + PregS vs. control) was significant at the 48 hour ($p=0.02$, Kruskal-Wallis test) and 7 day timepoints ($p=0.01$, Kruskal-Wallis test).

Reviewer 1 comment 2

The description of modeling component is difficult to follow and does not allow me to fully assess its value.

Reply

Regarding explanation of the modelling

We appreciate that the modelling in its complexity is difficult to follow, especially given the other aspects of the same manuscript. We have made the following changes to signpost the reader, and hopefully aid understanding of the modelling.

In the Results

p7, lines 25-27: We tested whether the *in vitro* changes in single neuron behaviour associated with NMDAR-Ab contributed to *in vivo* (interictal) EEG patterns by using a computational model of microcircuit dynamics. **Specifically, we tested the hypothesis that the microscale recordings can inform models of LFP dynamics using Bayesian model comparison of models with, and without prior information derived from the microscale. [...]**

p8, lines 27-29 and p9 line 1: **In order to assess whether the two main effects of interest (changes caused by NMDAR-Ab, and changes induced by seizures) can be separated in terms of their associated parameter changes, we performed a principal component analysis over three vectors containing the**

parameters fitted to control-Ab, NMDAR-Ab (interictal), and NMDAR-Ab (ictal) states. [...]

p9, lines 13-16: **Our hypothesis was that NMDAR-Ab alter brain dynamics in a way that makes neuronal circuits more susceptible to potentially seizure-inducing fluctuations in synaptic parameters. We tested this using the neuronal microcircuit models in simulation mode, allowing us to test any combination of parameter values, including mixtures of the three empirically fitted conditions. Specifically, we ran simulations across an ‘epileptogenesis’ x ‘ictogenesis’ parameter space.**

Reviewer 1 comment 3

For instance, the model results are based on time and frequency domain properties of the simulated data, but sample traces and spectra of these are not shown. In a combined approach used here, I would expect to see recorded data (such as depicted in the previous figures) and simulation results side-by-side. A direct comparison and similarity between recorded and simulated data would be convincing since Friston’s procedure to select the best amongst different simulations is very useful but won’t guarantee that the best result is appropriate.

Reply

Regarding model fits

It is absolutely correct that the model should reproduce the data. In fact, we are showing this *model fit* in Figure 5a – here the model fits align so closely with the empirical recordings, that they are in fact overlapping completely. We have clarified this as follows in the Results:

p8, lines 4-5: We then fitted a four population neural mass model [...] **with model fits for the power spectral densities shown for the final, selected model in Figure 5a.**

In the Figure 5 Legend

p44, line 5: [...] **a. Power spectra of LFP recordings of control-Ab injected mice, and interictal and ictal states of NMDAR-Ab injected mice are fitted with neural mass models of neuronal microcircuits. Dashed lines represent the model fit; solid lines represent the empirically measured spectra, **indicating faithful reproduction of empirically measured spectral features in the fitted dynamic causal model.****

And in the Methods:

P25, line 25: To be able to infer synaptic coupling parameters underlying the local field potential data, we extracted data from different relevant conditions and states (i.e. interictal control-Ab, interictal NMDAR-Ab, and ictal NMDAR-Ab) to then fit the microcircuit models using DCM to the **average power spectral densities of each of these conditions.**

Reviewer 1 comment 4

Furthermore, two principal components (Fig. 4e), using PCA, are interpreted as epileptogenesis and ictogenesis axes. The basis for this interpretation is unclear, including the results that build on this distinction (Fig. 5 and associated text) and the first paragraph in the Discussion section.

Reply

Regarding the axes for low-dimensional representations of the parameter changes

We have used the opportunity to clarify this in the manuscript. The PCA analysis motivates the subsequent analysis only in that the two main effects are separable and can thus be analysed in a meaningful way in a two dimensional reduced parameter space. We have added the following to the Results:

p8, lines 27-29 and p9 lines 1-11: **In order to assess whether the two main effects of interest (changes caused by NMDAR-Ab, and changes induced by seizures) can be separated in terms of their associated parameter changes, we performed a principal component analysis over three vectors containing the parameters fitted to control-Ab, NMDAR-Ab (interictal), and NMDAR-Ab (ictal) states. Projecting the parameter combinations for each of the three states onto the first two principal components (Figure 4e) demonstrates separability of the two effects with near orthogonal effects of NMDAR-Ab and seizure onset. For subsequent analyses, we therefore considered these effects independently: (i) The synaptic parameter difference derived from *control-Ab* and *NMDAR-Ab (interictal)* conditions captures the main effect of NMDAR-Ab infusion, which we know from the preceding results in a higher seizure likelihood, and is thus epileptogenic in the animals (hence, ‘epileptogenesis parameters’ for brevity). (ii) The synaptic parameter difference derived from the *NMDAR-Ab (interictal)*, and the *NMDAR-Ab (ictal)* conditions captures the main effect of transition into the seizure state, and thus represents ictogenesis (hence, ‘ictogenesis parameters’ for brevity).**

As well as

p9, lines 13-23 : **Our hypothesis was that NMDAR-Ab alter brain dynamics in a way that makes neuronal circuits more susceptible to potentially seizure-inducing fluctuations in synaptic parameters. We tested this using the neuronal microcircuit models in simulation mode, allowing us to test any combination of parameter values, including mixtures of the three empirically fitted conditions. Specifically, we ran simulations across an ‘epileptogenesis’ x ‘ictogenesis’ parameter space. For this analysis, we consider the parameter differences between *control* and *NMDAR-Ab (interictal)* condition the ‘epileptogenesis’ parameters since they capture the difference between brain dynamics in the non-epileptic control animals, and the animals with NMDAR-Ab induced spontaneous recurrent seizures. Similarly, we consider the parameter differences between *NMDAR-Ab (interictal)* and *NMDAR (ictal)* the ‘ictogenesis’ parameters since they capture the difference between brain dynamics in the interictal, and ictal state.**

And in the Discussion:

p12, lines 5-10: **We fitted synaptic parameters of microcircuit models to spectral features of LFP data in different experimental conditions corresponding to a non-epileptic, interictal condition (control), an epileptic and interictal condition (NMDAR-Ab interictal) and an epileptic and ictal condition (NMDAR-ictal). This allowed the separation of two main effects – epileptogenesis, and ictogenesis. In terms of parameter space, these effects were separable, meaning that they acted on different subsets of synaptic parameters.**

Reviewer 1 additional comments

Details

- The title does not reflect the combined in vitro, in vivo and in silico study.

Title reworded:

In vivo, in vitro and in silico electrophysiological study and neurosteroid treatment of an N-Methyl-D-Aspartate receptor antibody-mediated seizure model

- What is the oscillation in the 2nd trace in Fig. 3a?

The 2nd trace is an example of a theta oscillation in an awake rat infused with control antibodies. This has been removed from Figure 3a as it is not clear (Page 42);

- P.10, 2nd paragraph, line 5 and p.31: How were the domain borders for the parameters in Table 1 established?

We have included the following in the Methods:

p24, lines 27-28 and p25, lines 1-7: **Initial parameter values were taken from the standard formulation of the canonical microcircuit in dynamic causal modelling originally derived from cat visual cortex recordings (27, 83). These were adapted to this particular experiment in a two stage approach – posterior expected values from an initial model fit to the baseline data were subsequently used as priors in the subsequently fitted conditions. These priors do not define absolute limits to the parameter ranges, but restrict excessive between-condition differences through Bayesian integration of prior estimates (‘the priors’) and empirical likelihoods (‘the data’) to generate posterior densities (‘parameter estimates’) (90).**

- Fig. 4a only shows the effect on excitatory cells and omits the effect on the inhibitory ones.

These were informed by the microscale recordings presented in Figure 2. We did not model the non-significant effects on IPSCs

- I find that the Fig. 4b title ‘Microcircuit model’ and the diagram are misleading because this presentation suggests that a motif of individual neurons is modeled, while the model is a neural mass model

We have added references to ‘population models’ throughout the manuscript to clarify. However, we note that referring to connected neural masses as microcircuit and similar visualisations of neural masses to ours are very common and have a long history in specific subfields of computational neuroscience (e.g. Bastos et al., [Ref number 27]; Pinotsis et al., (www.frontiersin.org/articles/10.3389/fncom.2014.00149/full); Rosch et al., [Ref number 21]). We hope the additional reference to population models clarifies this particular aspect.

- The procedure used to get to Fig. 4c,d is hard to follow.

We have added the following explanation in the Results:

p8, lines 8-14: **We then performed Bayesian model comparison of these models using a free energy approximation of the model evidence – briefly, we fitted models where different subsets of parameters were informed by the microscale priors to the same data. By comparing the model evidence across these model inversions, we can thus evaluate whether the microscale priors allow a more parsimonious explanation for the LFP data that we have fitted. Given the uneven distribution of NMDAR in the cortical layers and across different neuronal populations, we also compare models where priors are applied only to different subsets of parameters, resulting in the model space of 22 models**

- P.29: What filter settings and sample rate were used for EEG acquisition?

We have added the following to the Methods (Page 30 line 18):

p23, lines 22-23: **...wide band pass 0.2-160Hz sampled at 512 samples per second.**

- Discussion: Paradoxical effects of reduced synaptic activity, including excitatory synapses, have been reported previously (e.g., van Drongelen et al., IEEE Trans Neural Syst Rehabil Eng. 2005 Jun;13:236; Pumain et al., Exp Neurol. 1985 89:250; ...). It would be useful to reference and briefly discuss these.

We have added the following to the Discussion section “Reductions in excitation can increase seizure propensity” with the suggested references:

p13, lines 16-28, and p14 lines 1-4: Similarly, recent work in a rat model of chronic temporal lobe epilepsy (TLE) demonstrates that dysfunctional regulation and resultant hypofunction of GluA (AMPA) subunits is an important factor in epileptogenesis (45). Defects in another AMPAR subunit, GluA2, are also found in neurodevelopmental disorders such as epileptic encephalopathy in children (46). **Previous experiments have also demonstrated that the weakening of excitatory synapses has a paradoxical effect on neural networks, with one early *in vitro* and computational study showing use of AMPAR blockers (CNQX) at moderate concentrations evoked seizure-like activity (47) and, more recently, anti-AMPA antibodies being demonstrated to produce weakened excitatory and inhibitory transmission and increased intrinsic excitability (48) similar to the effects demonstrated here with NMDAR-Abs. In fact, early experiments showed epileptic seizures can be sustained even when measured extracellular calcium reached levels that would completely block chemical synaptic transmission (49) highlighting the role of nonsynaptic neural network properties in epileptogenesis. Finally, as the Turrigiano laboratory has shown, synaptic scaling of excitatory amino acid receptors is co-regulated alongside intrinsic excitability (48) through the actions of CaMKIV, but GABA inhibition is separately regulated, emphasising the interdependence of synaptic, cellular and circuit mechanisms in control of network excitability (50).**

As well as modifying the following paragraphs for clarity:

p14, line 6, and lines 23-25: **That NMDAR hypofunction is proconvulsant has been shown though a variety of approaches.** Genetically modified mice lacking NMDARs in CA3 pyramidal neurons were more susceptible to kainate-induced seizures, and pharmacological blockade of CA3 NMDAR in adult wild-type mice produced similar results (51). Loss of function mutations in both the GluN1 and GluN2A subunits of the NMDAR have been identified in patients with epilepsy and neurodevelopmental disorders (52, 53). A recent dynamic causal modelling study of NMDAR-Ab encephalitis patient EEGs also showed that the deficit in signaling at NMDARs in patients with NMDAR-Ab encephalitis was predominantly located at the NMDA receptors of *excitatory* neurons rather than at inhibitory interneurons (54), as shown here in our model. By combining detailed multiscale electrophysiology and computational modelling, we provide further evidence that the direct synaptic effects of NMDAR-Ab identified from whole-cell patch clamp recordings offer a quantitative explanation for the epileptic brain dynamics captured *in vivo* using local field potentials.

In our circuit models of NMDAR-Ab-induced epileptogenesis, the parameter changes required to optimally model the associated LFP changes required not only changes in NMDAR-transmission related parameters, but also more widespread changes involving both excitatory and inhibitory neuronal populations. **This would appear to fit with the co-dependent changes seen in our rodent model and those of others as reported above. In addition to such compensatory and maladaptive network changes,** heterogeneity of NMDAR distribution at normal baseline function may also account for the unexpected effect of NMDAR-hypofunction in neuronal circuit dynamics. Selective forebrain pyramidal neuron NR1 knockout mice demonstrated both socio-cognitive deficits and *increased* pyramidal cell excitability (55), potentially driven by a release from inhibition by reducing putative NMDAR-mediated excitatory driving of inhibitory neuron activity, or by changing phase relationships between interacting neuronal populations. Similarly, hippocampal CA3 NMDA receptors may normally act to suppress the excitability of the recurrent network by restricting synchronous firing of CA3 neurons (51), with NMDAR-blockade releasing them from this suppression. Our modelling also suggests that NMDAR-Ab hypofunction may have stronger impacts on specific compartments of the affected microcircuitry.

Reviewer #2 (Remarks to the Author):

In this paper, the authors generated an NMDAR antibody induced model of encephalitis in rat and characterized seizure activities in vitro and in vivo. They further applied electrophysiological recording to show a reduced excitatory synaptic activity but no change in inhibitory activity, together with epileptiform activities in field potential recording. Modeling study suggests alterations in dynamic behavior of neural circuits may provide a mechanistic explanation. Interestingly, treatment with pregnenolone sulphate in vitro in both rodent and human slices resulted in a reduction of ictal activity, providing a potential treatment strategy through increasing NMDA receptors. While the characterization of the model is not new, the work does provide mechanistic insight about this disease and potential novel treatment methods. The study was well-designed and conclusions are supported, however, a few issues need to be addressed.

Reply

We thank the Reviewer for their positive comments on the manuscript.

Reviewer 2 comment 1

One major issue is that the authors did not assess and present data on the excitability and intrinsic properties of the recorded CA3 pyramidal neurons (e.g. potential changes in injected current-response curve, resting membrane potential, input resistance, etc). It is possible that hypoexcitability of the excitatory synaptic transmission may cause compensatory changes in neuronal intrinsic properties and excitability, making the neurons hyperexcitable. The paper does not have a direct measurement of related parameters. Without data for neuronal excitability and intrinsic properties, it is difficult to conclude that the network is hypoexcitable and not possible to know the contribution of neuronal intrinsic excitability.

Reply

We thank the Reviewer for this valuable advice and suggestion. These experiments have now been performed and results added to Figure 2: m-p are the new results added to the original Figure which was a-l (Pages 40-41);

a spontaneous EPSCs

d

g spontaneous IPSCs

j

m intrinsic cell excitability

o

Figure 2. Whole-cell patch-clamp recordings from hippocampal CA3 pyramidal cells *in vitro* 48 hours after intracerebroventricular injection of NMDAR-Abs show a reduction in excitatory but not inhibitory synaptic neurotransmission and intrinsic hyperexcitability. **a** Representative spontaneous EPSC (sEPSC) whole-cell patch clamp recordings 48 hours after injection with NMDAR-Abs (lower trace) or control Abs (upper trace). Scale bar 10pA vs. 5s. **b** The interevent interval (IEI) of sEPSCs recorded from CA3 pyramidal cells in NMDAR-Ab injected rodent slices (n=18 cells) compared to control Ab (n=8 cells) injected slices (p=0.003, Mann-Whitney). **c** The amplitude of the sEPSCs in CA3 pyramidal cells recorded from NMDAR-Ab injected slices (n=18) compared to controls (n=8) (p=0.04; Mann-Whitney). **d** Representative average sEPSCs recorded from NMDAR-Ab injected slices as compared to control-Ab injected slices. Scale bar 10pA vs. 10ms. **e** Cumulative frequency plot showing reduced decay time of sEPSCs recorded from NMDAR-Ab slices as compared to the controls (n=7 cells in each group). **f** Decay time of the sEPSCs in the NMDAR-Ab injected rodent slices as compared to the controls (n= 7 cells in each group; p= <0.001, Mann-Whitney). **g** Representative whole-cell patch clamp recording of spontaneous IPSC (sIPSC) from CA3 pyramidal cells after injection of NMDAR- or control-Abs. Scale bar 10pA vs. 5s. **h** The interevent interval of the sIPSCs recorded from CA3 pyramidal cells in control and NMDAR-Ab injected slices (n=6 cells in each group; p=ns, Mann-Whitney). **i** The amplitude of sIPSCs recorded from the control and NMDAR-Ab injected slices (n=6 cells in each group; p=ns, Mann-Whitney). **j** Representative average sIPSCs recorded from NMDAR-Ab injected slices as compared to control-Ab injected slices. Scale bar 5pA vs. 10ms. **k** The interevent interval of miniature IPSCs recorded (in the presence of TTX 1 μ M) from pyramidal cells in CA3 in control and NMDAR-Ab injected slices (n=6 cells in each group; p=ns, Mann-Whitney). **l** The amplitude of miniature IPSCs recorded (in the presence of TTX 1 μ M) from pyramidal cells in CA3 in control and NMDAR-Ab injected slices (n=6 cells in each group; p=ns, Mann-Whitney). **m** **Representative traces of CA3 pyramidal cell responses during depolarising current steps in CA3 pyramidal cells from control and NMDAR-Ab injected rodent slices.** **n** **Depolarising steps of different current intensities elicited significantly more spikes in the NMDAR-Ab injected rodent slices (n=15 cells) than in control-Ab (n=15 cells) treated slices (p=0.005, unpaired t-test).** **o** **The resting membrane potential was significantly depolarised in the NMDAR-Ab treated (n=14 cells) CA3 pyramidal cells compared to those treated with control-Ab (n=13 cells; p=<0.0001, Mann-Witney).** **p** **Input resistance was not significantly altered between the two conditions (NMDAR-Ab, n=14 cells vs. control-Ab n=15 cells; p=ns, Mann-Witney).** Measurements expressed as mean (M) \pm standard error of the mean (SEM).

And added to Results:

p6, lines 17-23: **Additionally, evaluation of the excitability and intrinsic properties of recorded CA3 pyramidal neurons showed them to be hyperexcitable (Figure 2 m-p).** Together these results indicate that local hippocampal networks are hyperexcitable when recorded *in vitro* 48 hours after a single ICV injection of NMDAR-Abs *in vivo*. **This circuit hyperexcitability is paradoxically associated with a reduction of excitatory neurotransmission in CA3 induced by the NMDAR-Abs with compensatory changes in the intrinsic properties of recorded CA3 pyramidal cells.**

Methods also amended:

p21, lines 5-14: **For intrinsic cell excitability experiments, whole-cell current-clamp recordings were made from hippocampal neurons using an Axopatch 700 series amplifier (Molecular Devices, USA) and with a K-gluconate-based intracellular solution containing (in mM) K-gluconate (120) hepes (40) KCL (10) Na₂ATP (2) Na-GTP (0.3) MgCl₂ (3) EGTA (0.5). Membrane potential was manually adjusted when required to -60mV by continuous somatic current injection. Membrane response and action potential firing were measured by applying depolarizing steps of current (duration of 1ms at 10pA increments). For spike/current intensity curves, the number of action potentials was plotted against the injected current. Input resistance was measured from the slope of a linear regression fit to the voltage-current plotted graph between -80pA to 0pA.**

Additionally, with the extra brain slices from these additional experiments we were able to use the tissue to add to the data presented in Supplementary Figure 1 g (Page 51; an additional data point added in Control-Ab column in 1g);

Supplementary Figure 1. Immunohistochemistry confirming NMDAR-Ab binding to juvenile Wistar rat hippocampus and morphology of pyramidal cells in CA3 used for whole-cell patch clamp recordings.

a Representative confocal image of hippocampus from sagittal brain slice prepared after acute ICV injection of monoclonal NMDAR-Ab SSM5 shows typical staining pattern with secondary anti-human IgG (green). Scale bar = 250 μ m. **b** Magnification of panel (a) shows the typical binding pattern of NMDAR-Abs with relative sparing of the granular cell layer (GCL) compared to the molecular cell layer (ML). Scale bar = 50 μ m. **c** Representative confocal image of hippocampus from sagittal brain slices prepared after chronic infusion of monoclonal NMDAR-Ab 12D7 after application of secondary anti-human IgG (green). Scale bar = 250 μ m. **d** Representative confocal image of hippocampus from sagittal brain slice prepared after chronic infusion of NMDAR-Ab positive IgG also shows typical staining pattern with secondary anti-human IgG (green). Scale bar = 250 μ m. **e** Representative confocal image of hippocampus from sagittal brain slices prepared after chronic infusion of healthy control IgG after application of secondary anti-human IgG (green). There is no specific binding of these antibodies. Scale bar = 250 μ m. **f** Representative confocal image of hippocampus from sagittal brain slices prepared after acute injection control monoclonal antibody mG053 after application of secondary anti-human IgG (green). There is no specific binding of these antibodies. Scale bar = 250 μ m. **g** The NMDAR-Ab treated brain slices showed increased median fluorescent intensity log EC₅₀ ratios (GCL vs. ML) compared to controls (control group contains: healthy control IgG n=2 animals, 6 brain slices and control monoclonal 12D7 n=3 animals, 9 brain slices; NMDAR group contains: SSM5 monoclonal n=8 animals, 22 brain slices, 003-102 n=2 animals, 2 brain slices and patient IgG n=3 animals 5 brain slices; p=0.035, Mann-Whitney). **h** Cells used for spontaneous excitatory and inhibitory current recordings were from putative pyramidal cells with the CA3 region; an example cell injected with neurobiotin shows clear features of a pyramidal cell (green, anti-streptavidin IgG). Scale bar 150 μ m.

We also amended Supplementary Table 1 to include additional animals used (Page 53);

Supplementary Table 1. Details of antibodies used in each experiment.

Procedure	Number of animals used with specified antibody					
	Healthy control IgG	Control monoclonal antibody mG053	Control monoclonal antibody 12D7	NMDAR monoclonal antibody SSM5	NMDAR monoclonal antibody 003-102	NMDAR -Ab patient IgG
ICV injection	N=5/8	-	N=3/8	N=11/14	-	N=3/14
Osmotic pump infusion with EEG transmitter	N=3/6	N=3/6	-	N=2/6	N=2/6	N=2/6

Reviewer 2 comment 2

Minor issue: The title of the paper may be modified to include electrophysiological findings, it seems interesting and novel.

Reply

Addressed above in Reviewer 1 additional comments.

Title reworded:

In vivo, in vitro and in silico electrophysiological study and neurosteroid treatment of an N-Methyl-D-Aspartate receptor antibody-mediated seizure model

REVIEWERS' COMMENTS:

Reviewer #1 (Remarks to the Author):

The authors did a thorough effort to address the critiques and modify the manuscript accordingly.

Reviewer #2 (Remarks to the Author):

My comments have been carefully addressed. The manuscript can be accepted for publishing.